# GRAPH SIGNAL PROCESSING MEETS MAMBA2: ADAPTIVE FILTER BANK VIA DELTA MODULATION

**Yehjin Shin**[*], **Seojin Kim**[*], **Noseong Park**[†]
KAIST
`{yehjin.shin, seojinkim, noseong}@kaist.ac.kr`

## ABSTRACT

State-space models (SSMs) offer efficient alternatives to attention with linear-time recurrence. Mamba2, a recent SSM-based language model, uses selective input gating and a multi-head structure, enabling parallel computation and strong benchmark performance. However, its multi-head recurrence operates independently without structured utilization or analysis. In this work, we propose a novel method called **H**ierarchical **AD**aptive filter bank for **E**fficient **S**SMs (*HADES*), a Graph Signal Processing (GSP)-inspired framework that reinterprets Mamba2 as an adaptive filter bank on a line graph. Our hierarchical architecture introduces two filter types: shared filters for global low-pass behavior and expert filters for local high-pass behavior, achieved through structured bias on the parameter $\Delta$. *HADES* achieves comparable performance to baseline models including Mamba2 across various benchmarks in language modeling, commonsense reasoning, and long-context retrieval, while using only **58.9%** of the original parameters. In this regard, *HADES* bridges GSP and neural sequence modeling, enabling efficient, hierarchical, and interpretable filtering within state-space models.

## 1 INTRODUCTION

Transformer architectures have emerged as the dominant approach for sequence modeling across a range of tasks, including text generation and machine translation. However, their inherent limitations, most notably the quadratic computational complexity, have motivated the development of more efficient, sub-quadratic alternatives (Gu et al., 2022; Yang et al., 2024b; Smith et al., 2023; Poli et al., 2023; Peng et al., 2023; Sun et al., 2024). In particular, Mamba (Gu & Dao, 2023) and Mamba2 (Dao & Gu, 2024) have demonstrated that continuous-time SSMs can match or surpass transformer baselines in diverse sequence modeling tasks.

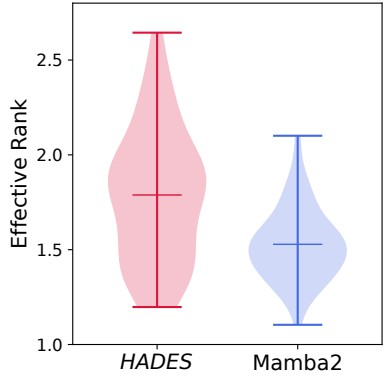

Figure 1: Distribution of layer-wise Effective Rank from the spectral responses of Mamba2 and *HADES*

Despite their empirical success, the internal structure of Mamba2—especially the role of multi-head recurrence—remains under-explored. Prior works have focused on improving long-context performance by delta modulation (Ben-Kish et al., 2025; Azizi et al., 2025; Ye et al., 2025). Another line of work (Wang et al., 2025) has critically examined the architectural limitations of SSMs and Mamba, highlighting issues such as recency bias and information bottlenecks. The authors proposed polarization as a potential solution to these challenges. However, their work mainly relies on simplified experimental settings, leaving its effectiveness on real-world tasks unexplored. Moreover, it is not well understood how different heads contribute to the overall representation, or whether they exhibit complimentary dynamics. In Fig. 1, effective rank analysis reveals that the heads learned by Mamba2 collapse into

---

[*]Equal contribution.
[†]Corresponding Author.

low-rank spectral subspaces, suggesting that most heads operate in highly overlapping frequency regimes rather than functioning as diverse, complementary filters (see Section 4.2 for details).

To address this issue, we reinterpret and enhance Mamba2 within the framework of Graph Signal Processing (GSP). Specifically, we model the input sequence as a signal on a line graph, where tokens serve as nodes and their temporal connections form edges. In this view, each recurrent head in Mamba2 functions as a graph filter applied to this signal. This perspective naturally leads to a filter bank interpretation, where individual heads can be understood as specialized filters, each capturing distinct spectral characteristics of the input.

Building on this formulation, we further propose a hierarchical filter bank model architecture, *HADES*, which allows adaptive and efficient information flow. *HADES* organizes filters into two functional categories: (1) *shared filters*, which perform globally consistent filtering across the sequence, and (2) *expert filters*, which adapt their filtering behavior on a per-token basis.

*HADES* demonstrates competitive performance across a diverse set of benchmarks, including two language modeling tasks, eight zero-shot commonsense reasoning tasks, and long-context retrieval, while utilizing only 58.9% of the parameters compared to Mamba2. For interpretability, we further examine *HADES* through case study and spectrum analysis, demonstrating how our filter bank approach affects the model's internal dynamics. By seamlessly integrating principles from GSP into neural sequence modeling, our method offers a scalable, hierarchical, and transparent filtering mechanism within state-space models.

Our main contributions are as follows:

- **GSP-Inspired Adaptive Filtering:** We establish a novel theoretical framework by reinterpreting Mamba2 as a graph filter bank operating over a line graph, creating a principled bridge between state-space models and GSP that enables more effective sequence modeling.

- **Hierarchical Filter Architecture:** We design an adaptive filtering system that optimally combines shared and expert filters through GSP-inspired delta modulation and bias mechanisms, enhancing model expressivity while maintaining computational efficiency.

- **Efficient and Scalable Performance:** Our approach achieves superior results across various benchmarks while using 58.9% of the parameters required by Mamba2. Through comprehensive spectral analysis, we demonstrate how our adaptive filtering strategy effectively captures both local and global dependencies in sequence data.

## 2 BACKGROUND

Our method, *HADES*, is based on a reinterpretation of structured sequence models from the perspective of Graph Signal Processing (GSP). We view the multi-head state-space model (SSM) as a learnable graph filter bank, where each head captures distinct frequency-selective dynamics. This section outlines the necessary background on SSMs and GSP to support this perspective.

### 2.1 STATE SPACE MODELS (SSMs) AND MAMBA

Structured state space models represent a new category of sequence models in deep learning, drawing connections to RNNs, CNNs, and traditional state space models. These models are motivated by a specific continuous system that processes a one-dimensional input sequence $x \in \mathbb{R}^T$ into an output sequence $y \in \mathbb{R}^T$ via an implicit latent state $h \in \mathbb{R}^{T \times N}$. Eq. 1 is a fundamental representation of organized SSMs.

$$
\begin{aligned}
h'(t) &= \bar{\mathbf{A}}h(t) + \bar{\mathbf{B}}x(t) \\
y(t) &= \mathbf{C}h(t) + \mathbf{D}x(t)
\end{aligned}
\qquad (1)
$$

$$
\begin{aligned}
h_t &= \mathbf{A}h_{t-1} + \mathbf{B}x_t \\
y_t &= \mathbf{C}h_t + \mathbf{D}x_t
\end{aligned}
\qquad (2)
$$

where $\mathbf{A}_t \in \mathbb{R}^{N \times N}$, $\mathbf{B}_t \in \mathbb{R}^{N \times 1}$, $\mathbf{C}_t \in \mathbb{R}^{1 \times N}$. This continuous SSMs in Eq. 1 are discretized to Eq. 2 through fixed formulas: $\mathbf{A} = f_A(\Delta, \bar{\mathbf{A}})$, $\mathbf{B} = f_B(\Delta, \bar{\mathbf{B}})$. For the remainder of this paper, we will omit the parameter $\mathbf{D}$ for exposition (or equivalently, assume $\mathbf{D} = 0$) because the term $\mathbf{D}x_t$ can

be viewed as a skip connection and is easy to compute.

$$\mathbf{K} = [\mathbf{CB}, \mathbf{CAB}, \dots, \mathbf{CA}^k\mathbf{B}]$$
$$y = x * \mathbf{K} \tag{3}$$

In S4 (Gu et al., 2022), the authors refer to this formulation as linear time-invariant (LTI), meaning the system parameters $\mathbf{A}, \mathbf{B}, \mathbf{C}$ do not change over time. The resulting sequence model can be computed either as a linear recurrence or as a global convolution using the kernel $\mathbf{K}$ in Eq. 3.

Using definitions from Dao & Gu (2024), we describe Mamba's internal dynamics. Each vector is designated as a row vector. Assuming that $U = [u_1, u_2, ..., u_T]^\top \in \mathbb{R}^{T \times d}$, that is, $u_i \in \mathbb{R}^d$, is a discrete time sequence of $T$ tokens, the inner equation for the $t$-th token of each head of the Mamba layer can be understood as follows:

$$h_t = \mathbf{A}_t h_{t-1} + \mathbf{B}_t x_t \in \mathbb{R}^{N \times P}, \quad y_t = \mathbf{C}_t h_t \in \mathbb{R}^P \tag{4}$$

$$o_t = W_o(\text{Norm}(y_t \odot W_z u_t)) \in \mathbb{R}^d \tag{5}$$

where $t$ is current time step, $x_t, y_t \in \mathbb{R}^P$ are projected input representation and output hidden representations of $t$-th token respectively, Norm denotes RMS normalization (Zhang & Sennrich, 2019), $W_z \in \mathbb{R}^{P \times d}$, $W_o \in \mathbb{R}^{d \times P}$ are trainable parameters. Especially, in Mamba2, $\mathbf{A}_t$ is scalar-identity matrix, i.e. $\mathbf{A}_t = a_t\mathbf{I}$. We denote $d$ for hidden representation dimension, $N$ for state size, $P$ for dimension of each head, $T$ for sequence length.

$$\Delta_{t,\text{base}} = W_\Delta u_t + b_\Delta \in \mathbb{R}, \quad \Delta_t = \text{Softplus}(\Delta_{t,\text{base}}) \in \mathbb{R} \tag{6}$$

By $\Delta$, Mamba implements input-dependent selection mechanism. $\Delta$ decides the discretization step size in Mamba, which is used to formulate SSM parameters $\mathbf{A}_t, \mathbf{B}_t$. Detailed parameterization of $\mathbf{A}_t, \mathbf{B}_t, \mathbf{C}_t, x_t$ are deferred to Appendix A.

## 2.2 GRAPH SIGNAL PROCESSING (GSP)

**Graph Signals and Filtering.** Graph Signal Processing (GSP) provides tools for analyzing and processing data defined over graph structures. In GSP, a signal is defined as a vector $\mathbf{x} \in \mathbb{R}^N$, where each element is associated with a node in a graph of $N$ nodes. One of the core operations in GSP is *graph filtering*, which can be viewed as a form of graph convolution. This operation emphasizes or suppresses specific frequency components of the signal based on the graph topology. Given a shift operator $\mathbf{S} \in \mathbb{R}^{N \times N}$—typically chosen as the adjacency matrix or the (normalized) graph Laplacian—a linear graph filter $\mathbf{G}$ is often defined as a polynomial in $\mathbf{S}$:

$$\mathbf{y} = \mathbf{G}\mathbf{x} = \sum_{k=0}^{K} h_k \mathbf{S}^k \mathbf{x}, \tag{7}$$

where $\mathbf{x}$ is the input graph signal, $h_k$ are the filter coefficients (also called filter taps), and $K$ is the filter order. This convolution operation aggregates information from neighboring nodes up to $K$ hops away, as determined by powers of the shift operator. This filtering can also be interpreted as a linear time-invariant (LTI) system on graphs, where the filter coefficients $h_k$ determine the system's impulse response under the graph structure. This system-theoretic view enables a conceptual connection to structured sequence models such as SSMs, which we explore in the following sections.

**Graph Filter Banks.** A graph filter bank applies multiple filters to a graph signal and combines their outputs to form a unified representation. Given a graph signal $\mathbf{x} \in \mathbb{R}^N$ and a graph shift operator $\mathbf{S} \in \mathbb{R}^{N \times N}$, the filter bank output can be expressed as:

$$\mathbf{y} = \mathbf{\Phi}\left(\left\{\mathbf{y}^{(i)}\right\}_{i=1}^{M}\right) = \mathbf{\Phi}\left(\left\{\sum_{k=0}^{K} h_k^{(i)} \mathbf{S}^k \mathbf{x}\right\}_{i=1}^{M}\right), \tag{8}$$

where $h_k^{(i)}$ are the coefficients of the $i$-th filter, $K$ is the filter order, $M$ is the number of filters in the bank, and $\Phi(\cdot)$ denotes the aggregation function over the filter outputs (e.g., concatenation, summation, or projection). This general form enables the system to capture diverse frequency characteristics of the graph signal through multiple learned filters. Our method adopts this perspective to reinterpret the multi-head SSM as a learnable graph filter bank, where each head corresponds to a distinct frequency response. While our model does not explicitly compute the graph spectrum, this filter bank perspective serves as a conceptual tool for understanding the role of the learned dynamic filters.

## 2.3 SSMs in the perspective of GSP

**SSMs as Graph Filters.** A one-dimensional token sequence can be naturally represented as a signal defined on a line graph (i.e., a linearly connected graph), where each token corresponds to a node and edges connect adjacent tokens in the sequence. This perspective enables the application of GSP tools to sequential data. In particular, the line graph admits a natural notion of convolution, where filtering operations over token sequences can be interpreted as graph convolutions. This provides a principled foundation for analyzing state-space models from a GSP perspective.

Specifically, the S4 model can be viewed as a LTI system operating on a line graph, where its kernel acts as the convolutional filter. This interpretation allows the SSM to be expressed as a graph convolution over the input sequence, offering a unified framework that bridges sequence modeling and GSP framework in Eq. 9:

$$\mathbf{y} = \mathbf{x} * \mathbf{K} = \sum_{k=0}^{K} \underbrace{(\mathbf{C}\mathbf{A}^k\mathbf{B})}_{h_k} \mathbf{S}^k\mathbf{x} \tag{9}$$

In contrast, Mamba can be interpreted as a linear time-varying (LTV) system operating on a line graph. Unlike an LTI system, which applies the same filter across all nodes, Mamba applies distinct, input-dependent filters at each node, enabling more flexible and adaptive sequence modeling. This formulation can be written as:

$$\mathbf{y}_t = \sum_{k=0}^{K} \underbrace{(\mathbf{C}_t\mathbf{A}_{t:t-k}\mathbf{B}_{t-k})}_{h_k^{(t)}} \mathbf{S}^k\mathbf{x}, \tag{10}$$

where $\mathbf{A}_{t:t-k} = \prod_{t-k}^{t} \mathbf{A}_i$ means cumulative product of $A_t$ from shift start index for $k$ hops. For more explanation on GSP and Mamba, refer to Appendix B.1.

**Multi-Head SSMs as Filter Banks.** Mamba2 employs multiple parameterized state-space recurrences, one per head, formulated as:

$$h_t^{(i)} = \mathbf{A}_t^{(i)} h_{t-1}^{(i)} + \mathbf{B}_t^{(i)} x_t, \quad y_t^{(i)} = \mathbf{C}_t^{(i)} h_t^{(i)}, \tag{11}$$

where $i \in [M]$ indexes the heads. This structure can be interpreted as a filter bank, with each head $i$ acting as a distinct filter applied to the input signal $x_t$.

$$\mathbf{y}_t^{(i)} = \Phi\left(\left\{\mathbf{y}_t^{(i)}\right\}_{i=1}^{M}\right) = \Phi\left(\left\{\sum_{k=0}^{K}(\mathbf{C}_t^{(i)}\mathbf{A}_{t:t-k}^{(i)}\mathbf{B}_{t-k}^{(i)})\mathbf{S}^k\mathbf{x}\right\}_{i=1}^{M}\right), \tag{12}$$

where $\mathbf{A}_i^{(j)} \in \mathbb{R}^{N \times N}, \mathbf{B}_i^{(j)} \in \mathbb{R}^{N \times 1}, \mathbf{C}_i^{(j)} \in \mathbb{R}^{1 \times N}$ are parameters of SSM equations and $M$ denotes the number of filters. Likewise, we can interpret multi-head architectures into a graph filter bank. In Fig. 2(a), we illustrate multi-head SSMs interpreted as graph filter banks.

Due to the use of head-specific recurrence parameters and potentially time-varying coefficients, the model is capable of exhibiting diverse temporal and spectral responses across heads. Nonetheless, Mamba2 imposes no explicit structural constraints or functional differentiation among heads. The learned filters are not directed toward specific frequency bands or contextual roles, resulting in an unstructured and static filter bank. This lack of coordination may hinder the model's ability to jointly capture both global and local dynamics in the input sequence. To address these limitations, we introduce a structured and adaptive filter bank design that encourages functional diversity across heads while enhancing the model's capacity to capture both global and local sequence.

## 3 Proposed Method

### 3.1 *HADES*: Hierarchical Adaptive Filter Bank for Efficient SSMs

From the perspective of node-adaptive filtering, a key challenge lies in how to effectively select and combine diverse filters. To enhance the structural expressivity of Mamba2 without compromising its efficiency, we propose an adaptive filter bank architecture based on GSP principles. Our approach decomposes the multi-head structure into two complementary components: shared filters and expert

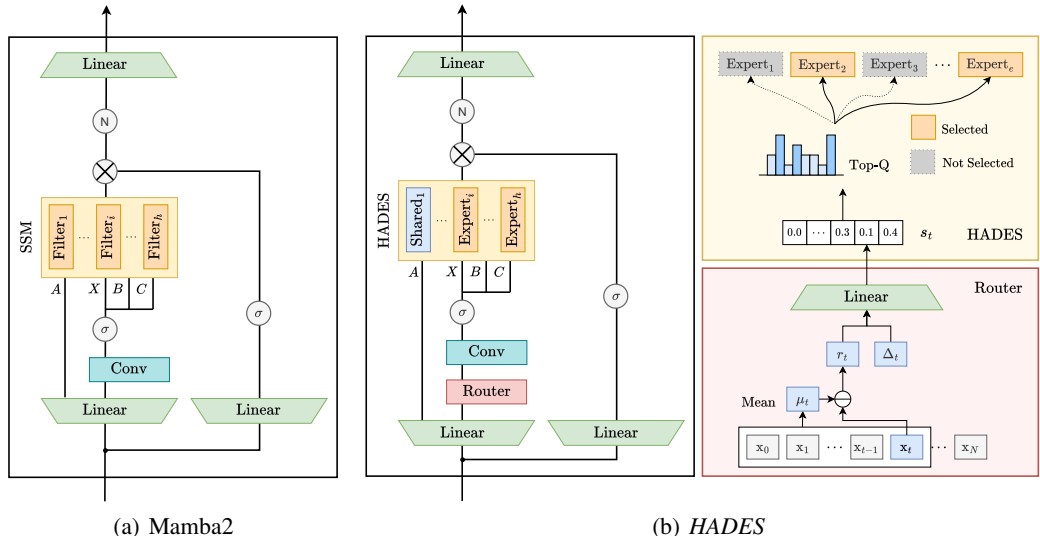

(a) Mamba2              (b) *HADES*

Figure 2: Architectural Comparison between Mamba2 and *HADES*. Mamba2 applies all filters uniformly to every input token, whereas *HADES* employs a routing mechanism that selects and activates filters conditioned on the spectral residual $r_t$ and $\Delta_t$.

filters. A router is employed to select the Top-Q expert filters, where the expert scores are computed based on the spectral residual and the characteristics of the input sequence.

Fig. 2 illustrates how our method functions as a filter bank. Fig. 2(a) shows the general Mamba2 architecture, where all filters are always utilized regardless of context. In contrast, the proposed method in Fig. 2(b), selects a subset of filters to be used at each timestep $t$. Among them, the shared filters are always applied, independent of the router's selection. This yields efficiency with fewer parameters than Mamba2; we defer analyses in Appendix F. In our method, from $M$ filters of the general Mamba2 architecture, $H$ filters are selected at each timestep. These $H$ filters are composed of $S$ shared filters and $E$ expert filters, which are dynamically chosen based on the routing mechanism. The final output is computed as a weighted linear combination of the selected filter outputs.

**Expert Filters.** To enable token-level adaptivity, we introduce a router that assigns a subset of expert filters to each token based on its frequency characteristics. Specifically, for each token at time step $t$, we compute the spectral residual as $r_t = x_t - \mu_t$, where $\mu_t$ is a running mean across the sequence, i.e., $\mu_t = \text{mean}(x_1, ..., x_t)$. The base delta parameter $\Delta_{t,\text{base}}$ is concatenated with the residual $r_t$, and the resulting vector is passed through a linear projection to compute selection scores $s_t$ for the expert filters:

$$s_t = f_e([\Delta_{t,\text{base}} \,\|\, r_t]), \quad r_t = x_t - \mu_t, \tag{13}$$

where $f_\text{e}$ is a function that computes expert selection scores based on both the base $\Delta_{t,\text{base}}$ and the token's spectral residual $r_t$, and $[\cdot \,\|\, \cdot]$ denotes vector concatenation. The resulting score vector $s_t \in \mathbb{R}^E$ contains a scalar score for each of the $E$ expert filters. The Top-Q filters with the highest scores are then selected and applied to the token. While expert filters are not explicitly assigned to specific frequency bands, their distinct $\Delta$ configurations induce varied update dynamics, implicitly shaping their responses based on the token's frequency characteristics.

The residual $r_t$ is used not only for expert selection, but also for modulating the delta value itself. Specifically, it introduces a frequency-sensitive bias that adjusts $\Delta$ in accordance with token-level frequency characteristics, spectral bias:

$$\Delta_{t,HADES} = \text{Softplus}(\Delta_{t,\text{base}} + \gamma \cdot f_b([\Delta_{t,\text{base}} \,\|\, r_t])) \tag{14}$$

where $f_\text{b}$ is a function that generates a content-aware adjustment to $\Delta_{t,\text{base}}$ based on the token's residual $r_t$, and $\gamma$ is a scaling hyperparameter that controls the strength of residual-based modulation. We use a single-layer linear projection for $f_\text{e}$ and $f_\text{b}$ in our implementation.

**Shared Filters.** Shared filters are always applied regardless of the router's selection, without additional bias and relying solely on the base $\Delta_{t,\text{base}}$. Designed to process globally smooth components

present throughout the sequence, they do not incorporate per-token modulation like expert filters, which explicitly respond to high-frequency deviations such as $x - \mu$. While not explicitly constructed as low-pass filters in the spectral domain, their uniform and content-agnostic operation tends to preserve low-frequency patterns and attenuate high-frequency variations. This behavior mirrors the role of fixed low-pass filters in GCN-style models (Dong et al., 2021) and structure-preserving averaging in (Wu et al., 2022), both of which apply smoothing without input-dependent bias. Such a design establishes a stable spectral foundation and reduces the risk of over-adaptation.

## 3.2 TRAINING LOSS TERMS

To ensure effective learning of the adaptive filter bank, it is crucial that the model utilizes a diverse set of filters rather than overfitting to a subset. Without appropriate regularization, the model may converge to using only a few filters, leaving others underutilized. Such underutilized filters may fail to generalize effectively, leading to underfitting, where some filters are insufficiently trained due to limited exposure to diverse sequence patterns. To address this challenge, we introduce a dual loss mechanism that encourages balanced filter utilization during training. Specifically, we apply two complementary objectives:

**Load Balance Loss.** To prevent the model from collapsing to a small subset of expert filters, we add a regularization term that encourages a more balanced usage of all available experts. Specifically, we compute the squared coefficient of variation over the selection scores to penalize high variance in expert preference:

$$\mathcal{L}_{\text{balance}} = \frac{\text{Var}(s_t)}{(\mathbb{E}[s_t])^2 + \epsilon} \tag{15}$$

where $s_t = f_e([\Delta_{t,\text{base}} \| r_t])$ is the vector of selection scores for the $E$ experts at time step $t$, and $\epsilon$ is a small constant for numerical stability. Minimizing this loss encourages the model to distribute its attention more uniformly across different experts.

**Diversity Loss.** This term ensures that each filter not only gets selected but also effectively contributes to the model's output. We achieve this by introducing a variance-based regularization on the filter responses, encouraging their outputs to be decorrelated. Concretely, we compute the pairwise similarity of normalized filter outputs and penalize deviations from orthogonality:

$$\mathcal{L}_{\text{diversity}} = \mathbb{E}_{i,j}\left[(\langle \hat{\mathbf{y}}_i, \hat{\mathbf{y}}_j \rangle - \delta_{ij})^2\right], \quad \delta_{ij} = \begin{cases} 1 & \text{if } i = j \\ 0 & \text{otherwise} \end{cases}, \tag{16}$$

where $\hat{\mathbf{y}}_i$ denotes the $\ell_2$-normalized output of the $i$-th expert filter. This loss encourages filter outputs to be mutually dissimilar, promoting specialization and functional diversity across experts.

**Final Loss Term.** The final training objective combines these two components:

$$\mathcal{L} = \mathcal{L}_{\text{task}} + \lambda_1 \cdot \mathcal{L}_{\text{balance}} + \lambda_2 \cdot \mathcal{L}_{\text{diversity}} \tag{17}$$

where $\mathcal{L}_{\text{task}}$ is the primary task loss (cross-entropy loss for language modeling), and $\lambda_1$, $\lambda_2$ are hyperparameters controlling the strength of the selection and diversity losses respectively. This dual loss mechanism ensures that the model effectively learns a diverse set of filters, each specialized for different aspects of the input sequence.

## 4 EMPIRICAL STUDIES

### 4.1 EVALUATION

**Setup.** Our experiments encompass a comprehensive comparison of recent state-of-the-art architectures. We evaluate against the following baselines: Linear Transformer (Katharopoulos et al., 2020), RetNet (Sun et al., 2024), Mamba (Gu & Dao, 2023), Mamba2 (Dao & Gu, 2024), and DeltaNet (Yang et al., 2024b). We trained all models using approximately 200B tokens from the Pile dataset (Gao et al., 2020). We adopt all baseline implementations from `flash-linear-attention` (Yang & Zhang, 2024). For our model, we used hyperparameter set of $H = 16$, $S = 8$, $\lambda_1 = 1e - 3$, $\lambda_2 = 1e - 3$, $\gamma = 25e - 2$. For ablation studies, we maintain all hyperparameter configurations constant, modifying only the target under evaluation. Detailed settings are in Appendix C.

Table 1: Performance comparison on language modeling and zero-shot common-sense reasoning. The best results are highlighted in **bold**, while the second-best results are underlined. Avg. denotes the average of accuracies and normalized accuracies over 8 tasks. With only 58.9% of parameters compared to baseline models, *HADES* achieves comparable or even better performance.

| Model | Train. ppl↓ | Wiki. ppl↓ | LMB. ppl↓ | LMB. acc↑ | BoolQ acc↑ | Hella. acc_n↑ | Wino. acc↑ | ARC-e acc↑ | ARC-c acc↑ | PIQA acc↑ | OBQA. acc_n↑ | Avg. 8 tasks↑ |
|---|---|---|---|---|---|---|---|---|---|---|---|---|
| Linear Transformer | 2.49 | 45.43 | 73.93 | 24.06 | 61.50 | 28.20 | 51.30 | 42.05 | 21.76 | 60.55 | 27.60 | 39.63 |
| RetNet | 2.41 | 34.12 | 29.46 | 35.36 | 55.57 | 31.31 | 51.70 | 44.49 | 23.46 | 62.40 | 28.00 | 41.54 |
| DeltaNet | 2.29 | 33.25 | 26.82 | 35.75 | 54.07 | 31.40 | 49.96 | 44.11 | 22.18 | 63.60 | **29.60** | 41.33 |
| Mamba1 | 2.53 | 47.51 | 85.53 | 22.43 | **62.17** | 28.71 | 50.67 | 42.09 | 22.35 | 60.72 | 26.60 | 39.47 |
| Mamba2 | 2.33 | **31.34** | 24.38 | 36.46 | 53.88 | 32.62 | 50.83 | **45.29** | **24.15** | 63.44 | 26.40 | 41.63 |
| *HADES* (Ours) | 2.31 | 31.48 | **21.74** | **39.24** | 58.84 | **32.82** | **52.64** | 45.03 | 22.01 | **63.93** | 28.80 | **42.91** |

**Language Modeling and Commonsense Reasoning.** In Table 1, we present the performance of each model across multiple benchmarks, including language modeling perplexity and zero-shot accuracy on commonsense reasoning benchmarks for models with 370M parameters. Even with **58.92%** of parameters (218M), *HADES* consistently outperforms other baselines, including Linear Transformer, RetNet, Mamba, Mamba2, and DeltaNet. More experiment results are in Appendix D.2.

**Long-context Retrieval.** To evaluate the long-range memory capacity, we adopt the passkey retrieval task, where a key-value pair is planted at various depth in a long sequence and queried at the end. Experimental details are in Appendix C.2. In Fig. 3, the results show that our model significantly outperforms Mamba2, demonstrating the effectiveness of our GSP-inspired adaptive filtering in retaining distant dependencies.

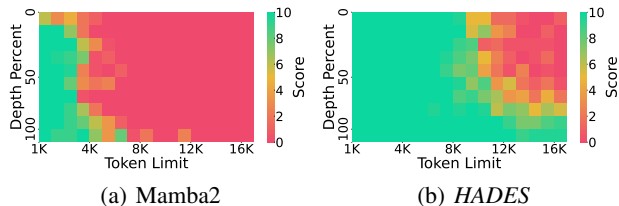

(a) Mamba2    (b) *HADES*

Figure 3: Passkey retrieval result of Mamba2 and *HADES*

## 4.2 IN-DEPTH ANALYSIS

**Why Graph Signal Processing is Effective?** Our method is fundamentally grounded in GSP, which offers a principled framework for understanding and designing filtering behavior over graph-structured signals. This spectral view brings clarity to the model's internal dynamics: shared filters serve as filters that capture smooth, global trends across the sequence, i.e., low frequency components, while expert filters, modulated by local signal variations, act as filters that adaptively detect sharp, localized changes, i.e., high frequency components, or occasionally choose to observe low frequency components. The GSP perspective bridges the gap between theoretical signal processing and practical neural architecture design. Our model is not just a collection of independent recurrent heads, but an adaptive filter bank, where each head (filter) can specialize in processing specific frequency bands and is complementary to each other. This structured filtering hierarchy aligns with the spectral nature of graph signals, while maintaining the scalability and efficiency of our model.

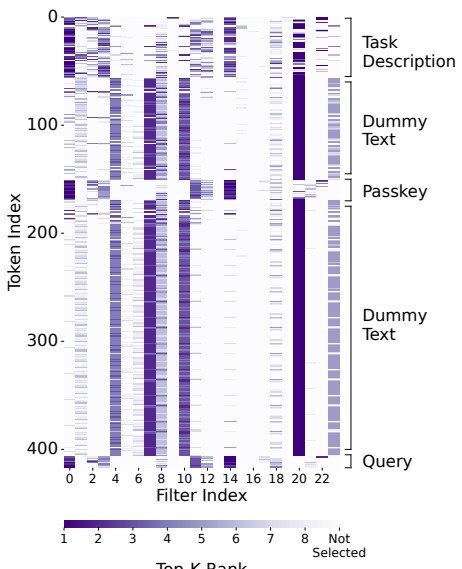

Figure 4: Expert filter selection in Passkey Retrieval task

**Filter Selection Analysis.** To investigate filter selection patterns of *HADES*, we analyze the selection tendencies of each filter under the passkey retrieval task, which provides a relatively clean separation of task-specific and task-irrelevant tokens. The prompt used for passkey retrieval is divided into four semantic regions: Task Description, Passkey, Query, and Dummy Text. The exact textual

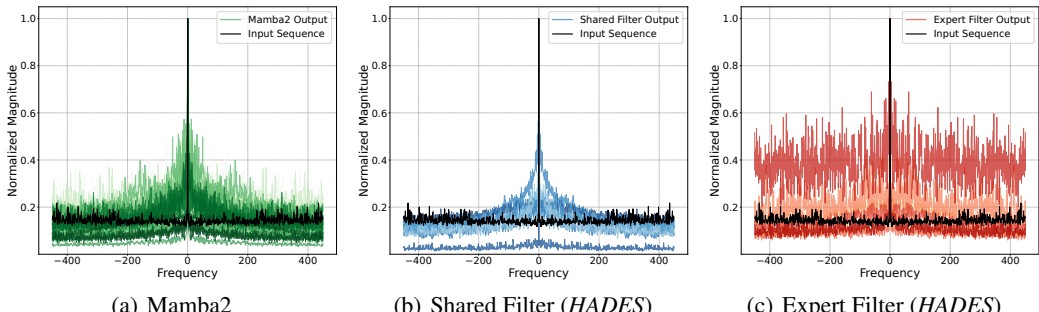

(a) Mamba2      (b) Shared Filter (*HADES*)      (c) Expert Filter (*HADES*)

Figure 5: Spectrum of filter inputs and outputs from Mamba2 and *HADES*. The x-axis represents the Fourier frequency bins, and the y-axis shows the normalized magnitude of the Fourier coefficients, with larger values indicating stronger frequency components (see Appendix E.4 for details).

contents of each part are provided in Appendix C.2. Among these, Passkey and Query are directly related to the task and thus can be considered as task-specific content. As shown in Fig. 4, we observe distinct selection patterns across filters depending on the regions of the input prompt. Filter 0, 11, and 14 are predominantly activated in task-specific regions, such as the Passkey and Query regions. In contrast, Filter 4, 7, and 20 are tend to be selected primarily within Dummy Text region, suggesting a possible specialization for irrelevant or noisy input. Notably, the Task Description region, located at the beginning of the prompt, shows broader diversity in filter selection compared to other parts. This behavior suggests that in the initial stage of the prompt, the model may leverage multiple filters to encode general context and task intent before narrowing down to more specialized filters in later segments. This transition from diverse to focused selection suggests an adaptive routing mechanism, where filters self-organize to capture both high-level instructions and low-level execution signals.

**Output Spectrum Analysis.** For output spectrum analysis, we apply Fourier transform to filter outputs obtained from a randomly sampled sentence from the Pile dataset. In Fig 5, the kernel characteristics directly influence the information each model learns; while Mamba2's outputs in Fig. 5(a) mainly preserve low-frequency information, *HADES* captures a more diverse range of signals through the shared and expert filters. The shared filters (Fig. 5(b)), shaped by smooth kernels, consistently emphasizing low-frequency components, aligning with their role in capturing stable, global information across the sequence. In contrast, expert filters in Fig. 5(c), learned by rippled kernels, demonstrate more dynamic filter response, highlighting their adaptive specialization in capturing localized details. This spectral distinction reflects our model's design: shared filters ensure a stable contextual foundation, while expert filters dynamically adapt to fine-grained variations.

**Filter Frequency Response Analysis.** To characterize the intrinsic spectral behavior of our model, we analyze the frequency response of its learned filters, examining how the kernels themselves react to different frequency components. For the filter frequency response, we analyzed the frequency response of its filters (cf. Eq. 12), following the procedure proposed in Wang et al. (2022). Our analysis in Fig. 6 reveals that the majority of Mamba2's learned filters behave as smooth kernels. This bias toward smooth filtering implies that Mamba2 tends to prioritize low-frequency or long-range information while insufficiently capturing high-frequency variations. As a result, many heads converge to similar, general-purpose behaviors, leading to substantial redundancy across filter outputs. A detailed analysis of this output redundancy is provided in Appendix E.3.

In contrast, *HADES* exhibits a more diverse set of filtering behaviors: alongside smooth kernels, we also observe clear rippled kernels that respond to higher-frequency components. The presence of ripple kernels provides enhanced sensitivity to high-frequency bands, and through our modulation and expert-

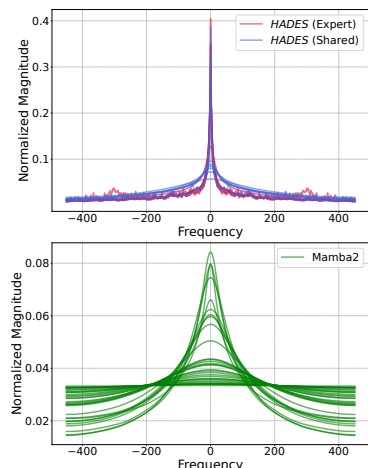

Figure 6: Filter Spectral Responses of Mamba2 and *HADES* (see Appendix E.5 for details).

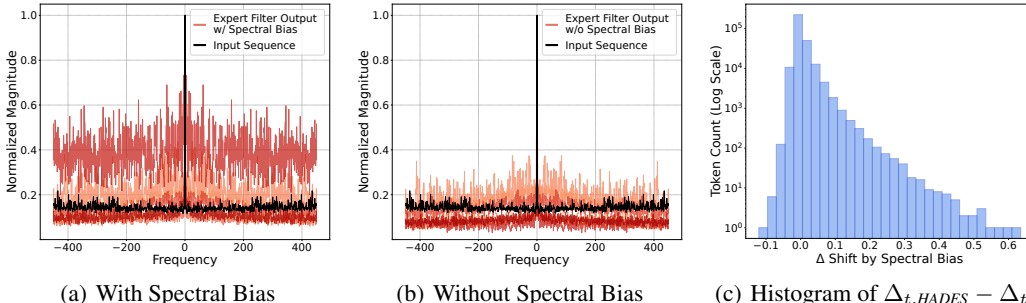

| (a) With Spectral Bias | (b) Without Spectral Bias | (c) Histogram of $\Delta_{t,HADES} - \Delta_t$ |

Figure 7: (a-b) Comparison on frequency spectrum of filter outputs from expert filter with bias and expert filter without spectral bias. The x-axis represents the Fourier frequency bins, and the y-axis shows the normalized magnitude of the Fourier coefficients, with larger values indicating stronger frequency components (see Appendix E.4 for details). (c) Histogram of spectral bias.

selection mechanisms, the model learns information at multiple spectral resolutions. As a result, *HADES* leverages not only low-frequency dominant global context but also high-frequency driven fine-grained structure in a more balanced manner. This indicates that our high-frequency–aware modulation and routing design enables the model to capture a wider range of spectral patterns. Consistent with this observation, *HADES* shows a noticeably higher effective rank than Mamba2, suggesting that it learns a more expressive and less redundant filter bank.

**Effect of Spectral Bias.** We further explore the impact of spectral bias on expert filters. Specifically, Fig. 7(b) shows the frequency spectrum of the output generated by using the original delta without spectral bias, $\Delta_t$. In contrast, Fig. 7(a) illustrates the effect of applying spectral bias to expert filters, $\Delta_{t,\text{HADES}}$, where a clear upward shift in frequency distribution is observed. This shift indicates that the delta values are tuned to capture higher-frequency details, enabling the model to learn finer-grained information. Fig. 7(c) presents a log-scale histogram of the difference between $\Delta_{t,\text{HADES}}$ and $\Delta_t$, calculated over 25 randomly sampled sentences from the Pile dataset, totaling approximately 38,000 tokens. Throughout our analysis, we observe that the spectral residual bias is generally positive, which encourages larger delta values and enables the model to effectively capture high-frequency information, which aligns with Fig. 7(a). Occasionally, the bias becomes negative, reducing the step size for certain tokens and allowing the model to better capture global context. This adaptive mechanism allows *HADES* to flexibly balance the extraction of local and global information, adjusting to the needs of each token in context.

## 4.3 ABLATION STUDIES

In this section, we conduct ablation studies on our model, *HADES*, to systematically evaluate the impact of each component and design choice on model performance. For more ablation result, refer to Appendix E.1. We first ablate on two auxiliary losses: $\mathcal{L}_{\text{diversity}}$ and $\mathcal{L}_{\text{balance}}$. As shown in Table 2, the best performance is achieved when both losses are applied together. Interestingly, without $\mathcal{L}_{\text{balance}}$, performance drops as filter selection becomes overly concentrated on a few filters, leaving the rarely selected filters under-trained and preventing them to learn dynamics effectively when they are eventually chosen. We also evaluate the impact of different filter configurations: Using Only Shared Filters and Using Only Expert Filters. Using only shared filters outperforms using only expert filters, as shared filters consistently capture global low-frequency information, while expert filters adaptively capturing low and high frequency information.

Table 2: Ablation Studies

| Methods | Wiki. ppl ↓ | LMB. ppl ↓ | Avg. 8 tasks ↑ |
|---|---|---|---|
| *HADES* (Ours) | **31.51** | **21.74** | **42.91** |
| w/o $\mathcal{L}_{\text{balance}}$ | 34.73 | 26.84 | 41.57 |
| w/o $\mathcal{L}_{\text{diversity}}$ | 33.83 | 27.40 | 42.15 |
| Only Shared Filters | 34.55 | 27.64 | 42.21 |
| Only Expert Filters | 36.34 | 30.12 | 41.68 |
| Fixed | 34.55 | 27.64 | 42.21 |
| Random | 35.78 | 32.77 | 41.03 |
| Pos. Bias | 30.23 | 21.93 | 42.15 |
| No Bias | 34.57 | 28.79 | 41.11 |

We conducted additional ablations on filter selection and delta modulation. Fixed uses a constant set of filters; and Random randomly samples filters without considering the input. We further

evaluated three variants in Appendix E.1. For delta modulation, Position Bias (Pos. Bias) adds token-wise positional information; and No Bias removes modulation entirely. In filter selection, Fixed outperformed random selection, and Input-only achieved reasonable performance. However, spectral-based selection (*HADES*) produced the best results by adapting to input characteristics. Pos. Bias helped, regarding delta modulation, but original bias was more effective, as it captures relative importance beyond absolute positions.

## 5 RELATED WORKS

**Graph Signal Processing in Language Modeling.**  Recent work has explored interpreting Transformer architectures through GSP. In this view, the self-attention mechanism functions as a graph filter, where the attention matrix acts as a learned adjacency matrix, with tokens as nodes and attention weights defining edges. GFSA (Choi et al., 2024) explicitly models self-attention as a graph filter on a fully connected graph, while ContraNorm (Guo et al., 2023) treats it as a normalized adjacency matrix, connecting it to Graph Neural Networks. The Anti-Oversmoothing framework (Wang et al., 2022) further characterizes self-attention as a low-pass filter, highlighting its smoothing effect in the spectral domain. Unlike Transformers, represented with fully connected graphs, SSMs operate sequentially, best represented by a line graph with unidirectional information flow. This insight motivates our GSP-based filter bank approach for Mamba2, a recent SSM-based model.

**Adaptive Filtering and Mixture of Experts.**  Recent methods have improved multi-head attention efficiency by dynamically selecting or weighting attention heads. Mixture of Attention Heads (MoA) (Zhang et al., 2022) treats each head as an independent expert, with a router dynamically selecting a subset of K heads per token, enhancing efficiency by focusing on the most relevant heads. Interpreted through a GSP lens, MoA functions as adaptive filtering, where tokens selectively activate the most suitable filters (heads). Building on this, Mixture-of-Heads Attention (MoH) (Jin et al., 2024) further advances this approach by using a router to assign weights to all heads, rather than selecting a subset. This allows each token to receive a weighted combination of all head outputs, offering greater flexibility. Unlike MoA, which treats heads independently, MoH uses a shared set of heads with adaptive weights, providing a more direct form of adaptive filtering where filter weights are continuously adjusted. Conventional Mixture-of-Experts (MoE) frameworks (Shazeer et al., 2017; Fedus et al., 2022; Dai et al., 2024) route tokens across multiple expert networks, expanding capacity at the cost of greater computation and memory. Our approach embodies the MoE principle of conditional computation in a lightweight form: routing remains within a single architecture, with each head functioning as a filter in a filter bank. In this sense, it is structurally closer to MoA/MoH, yet conceptually aligned with the adaptive utilization underlying MoE.

**Modulation of SSMs.**  Mamba's recursive state update leads to information loss as context length increases, a problem noted in various studies. DeciMamba (Ben-Kish et al., 2025) addresses this by measuring information loss using Effective Receptive Field (ERF) and removing less important tokens with low $\Delta$ values. MambaExtend (Azizi et al., 2025) improves on this by offering a training-free scaling method, adjusting $\Delta$ values directly to enhance long-context performance. LongMamba (Ye et al., 2025) further refines this by separating global and local channels, using token filtering in global channels to improve memory efficiency and extend the receptive field. Another work tries to emphasize polarization of $\mathbf{A}$, thereby allowing SSMs to capture vanishing influence of earlier tokens in long sequences (Wang et al., 2025).

## 6 CONCLUSION

In this work, we proposed a hierarchical adaptive filtering architecture for SSMs, bridging the gap between SSMs and GSP. Our method, *HADES*, reinterprets Mamba2 as a GSP-inspired filter bank, introducing a novel separation of shared and expert filters via delta modulation and spectral residual bias. This design enables efficient, frequency-adaptive filtering, significantly improving performance across language modeling, commonsense reasoning, and long-context tasks, while maintaining parameter usage at 58.9% of baseline models including Mamba2. Our approach not only advances the understanding of SSMs from a GSP perspective but also demonstrates the effectiveness of structured, adaptive filtering in neural sequence modeling. We leave limitations and future work in Appendix G.

ACKNOWLEDGMENTS

This research was supported by the MSIT (Ministry of Science, ICT), Korea, under the Global Research Support Program in the Digital Field program (RS-2024-00436680), Developing a Sustainable Collaborative Multi-modal Lifelong Learning Framework (RS-2022-II220113), and AI Research Hub Project (RS-2024-00457882) supervised by the IITP (Institute for Information & Communications Technology Planning & Evaluation). This project is supported by Microsoft Research Asia.

ETHICS STATEMENT

This work does not involve human participants, sensitive information, or proprietary datasets. All experiments are conducted on publicly available benchmarks such as the Pile. Although advances in long-context language modeling may broaden downstream applications, we recognize potential risks of misuse, including harmful or misleading content generation. Our contributions are purely methodological, and we encourage responsible use of this technology in accordance with the ICLR Code of Ethics.

LLM STATEMENT

In preparing this manuscript, we used a large language model (LLM) solely to refine the language and presentation. The tool was applied to enhance readability and polish the text, without influencing the conceptual development or technical contributions of the work. The responsibility for the research content and findings remains entirely with the authors.

REPRODUCIBILITY STATEMENT

Our implementation is publicly available at `https://github.com/yehjin-shin/HADES`. It contains source code, evaluation scripts, and configuration files for all reported experiments. All datasets used in this study are publicly available. Experimental setups and detailed hyperparameters are documented in Appendix C.

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

# A  FULL MAMBA2 ARCHITECTURE

Given an input sequence $U = [u_1, u_2, ..., u_T]^\top \in \mathrm{R}^{T \times d}$, a Mamba2 block with $d$ channels is built on top of the S6 layer via the following formula, generating output sequence $O = [o_1, o_2, ..., o_T]^\top \in \mathbb{R}^{T \times d}$:

$$h_t = \mathbf{A}_t h_{t-1} + \mathbf{B}_t x_t \in \mathbb{R}^{N \times P}, \quad y_t = \mathbf{C}_t h_t \in \mathbb{R}^P \tag{18}$$

$$o_t = W_o(\mathrm{Norm}(y_t \odot W_z u_t)) \in \mathbb{R}^d \tag{19}$$

where $W_x, W_z \in \mathrm{R}^{d \times P}$, $W_o \in \mathbb{R}^{P \times d}$ are trainable parameters. Each Mamba2 block consists of $M$ heads, so that $M \times P = d$ , which are computed in parallel, the result of which is summed together. We can specify how each matrices are created for each head:

$$
\begin{aligned}
\bar{\mathbf{A}}_t &= a_t \mathbf{I} \in \mathbb{R}^{N \times N} \\
a_t &= \exp(-\Delta_t \exp(A)) \in \mathbb{R} \\
\mathbf{B}_t &= \Delta_t \bar{\mathbf{B}}_t \in \mathbb{R}^{N \times 1} \\
\bar{\mathbf{B}}_t &= \sigma(\mathrm{Conv}(W_B u_t)) \in \mathbb{R}^{N \times 1}
\end{aligned}
\tag{20}
$$

$$
\begin{aligned}
\mathbf{C}_t &= \sigma(\mathrm{Conv}(W_C u_t))^\top \in \mathbb{R}^{1 \times N} \\
\Delta_{t,\mathrm{base}} &= W_\Delta u_t + b_\Delta \in \mathbb{R} \\
\Delta_t &= \mathrm{Softplus}(\Delta_{t,\mathrm{base}}) \in \mathbb{R} \\
x_t &= \sigma(\mathrm{Conv}(W_x u_t)) \in \mathbb{R}^{P \times 1}
\end{aligned}
\tag{21}
$$

where $W_B, W_C \in \mathbb{R}^{N \times d}$, $W_\Delta \in \mathbb{R}^{1 \times d}$. $\sigma$ denotes SiLU activation function and $\mathrm{Conv}(\cdot)$ denotes a channel-wise one-dimensional convolution. By $\Delta$, Mamba2 implements input-dependent selection mechanism. $A_t$ performs as decay-ratio as it is cumulatively multiplied. DeciMamba (Ben-Kish et al., 2025) elaborates the condition of $a_t$. For computational stability, $\Delta > 0$ and $A < 0$ is guaranteed in original implementation. Therefore, we can conclude $a_t \in (0, 1)$.

Using this Mamba2 block, we can derive layer-wise Mamba2 architecture with $L$ layers as below. For initial input, input sequence is $I = [i_0, i_1, ..., i_T] \in \mathbb{R}^T$ where $i_t \in [V]$ and we have $U^{(l-1)} = [u_1^{(l-1)}, u_2^{(l-1)}, ...u_T^{(l-1)}]$ as input sequence for the $l$-th layer. $O^{(l)} = [o_1^{(l)}, o_2^{(l)}, ...o_T^{(l)}]$ serves as output sequence of $l$-th Mamba2 layer $\mathrm{Mamba}^{(l)}$, $V$ denotes vocab size and $P \in \mathbb{R}^{T \times V}$ denotes final logits.

$$U^{(0)} = \mathrm{Embedding}_{in}(I) \in \mathbb{R}^{T \times d} \tag{22}$$

$$O^{(l)} = \mathrm{Mamba}^{(l)}(\mathrm{Norm}[U^{(l-1)}]) \in \mathbb{R}^{T \times d} \tag{23}$$

$$P = \mathrm{Embedding}_{out}(\mathrm{Norm}([O^{(L)}]) \in \mathbb{R}^{T \times V} \tag{24}$$

Here, the output of the $l$-th layer is used as the input for the $l + 1$-th layer, i.e. $O^{(l)} = U^{(l)}$.

# B  MORE DISCUSSION

## B.1  DETAILED EXPLANATION ON *HADES*

**Detailed explanation on connection of GSP and *HADES*.** While it is true that most GSP-based analyses traditionally assume time-invariant dynamics (e.g., S4), we emphasize that GSP can indeed be extended to handle time-variant systems. In Eq. 10, we explicitly formulate Mamba as a Linear Time-Variant (LTV) system operating over a line graph. From the GSP perspective, this corresponds to a Node-Variant Graph Filter (NVGF) (Gama et al., 2022), where each node (i.e., each time step in the sequence) is associated with its own filter coefficients, modulated by input-dependent dynamics via the state-space parameters $\mathbf{A}, \mathbf{B}, \mathbf{C}$.

In the node-invariant case, corresponding to an LTI system such as S4, the output can be expressed as

$$\mathbf{y} = \sum_{k=0}^{K} h_k \mathbf{S}^k \mathbf{x}, \tag{25}$$

with coefficients given in S4 by $h_k = \mathbf{C}\mathbf{A}^k\mathbf{B}$. By contrast, in the node-variant case, which characterizes LTV systems such as Mamba, the filtering operation becomes

$$\mathbf{y} = \sum_{k=0}^{K} \mathrm{diag}(\mathbf{h}_k) \, \mathbf{S}^k \mathbf{x}, \tag{26}$$

where each $\mathbf{h}_k = (\mathbf{h}_k^{(0)}, \mathbf{h}_k^{(1)}, \cdots, \mathbf{h}_k^{(N-1)}) \in \mathbb{R}^N$ assigns distinct filter values to each node. For Mamba in particular, these coefficients satisfy

$$\mathbf{h}_k^{(t)} = \mathbf{C}_t \mathbf{A}_{t:t-k} \mathbf{B}_{t-k}. \tag{27}$$

This formulation is grounded in established work on NVGF in GNNs [1], and our approach uniquely applies these principles to 1D sequences via the line graph interpretation. Thus, our work extends GSP tools to LTV systems like Mamba, offering a principled and novel perspective that complements traditional LTI analyses such as those for S4.

Mamba can also be understood as a form of linear attention. Conversely, linear attention itself can be reformulated using the state-space model equations employed in Mamba (Dao & Gu, 2024; Sieber et al., 2024). For Mamba viewed as a Linear Time-Variant (LTV) system, the state-space equations are given by

$$h_i = A_i h_{i-1} + B_i u_i, \qquad y_i = C_i h_i + D_i u_i. \tag{28}$$

In the case of linear attention, these parameters can be expressed in the same form, with

$$A_i = \frac{(\mathbf{elu}(q_{i-1}) + 1) \sum_{j=0}^{i} (\mathbf{elu}(k_j) + 1)}{(\mathbf{elu}(q_i) + 1) \sum_{j=0}^{i} (\mathbf{elu}(k_j) + 1)}, \tag{29}$$

$$B_i = \frac{1}{(\mathbf{elu}(q_{i-1}) + 1) \sum_{j=0}^{i} (\mathbf{elu}(k_j) + 1)} \mathbb{I}_d \times (\mathbf{elu}(k_j) + 1), \tag{30}$$

$$C_i = \mathbb{I}_d \times (\mathbf{elu}(q_i) + 1). \tag{31}$$

Under this formulation, linear attention can also be interpreted in Mamba's equation form. Ultimately, both models can be framed as linear time-variant systems within the context of GSP, offering a unified analytical view of these seemingly distinct architectures.

**Pseudo Code.** In this paragraph, we display pseudo-code for our method on prefill and decode stage respectively.

```
1  def forward(u, seqlen=None, seq_idx=None, cu_seqlens=None, inference_params=None):
2      # Check for inference cache
3      if inference_params exists:
4          update cache params
5          go to decode function
6
7      # 1. Linear projection of input
8      zxbcdt = in_proj(u)
9      zxbc, dt = split(zxbcdt into features and dt terms)
10
11     # 2. Compute spectral residual
12     spectral_residual = u - cumulative_mean(u)
13     udt = concat(spectral_residual, dt)
14
15     # 3. Project for routing
16     hb = h_proj(udt)
17     h, spectral_bias = split(hb into scores and bias terms)
18
19     # 4. MoE routing
20     select_ids = topk(h)
21     shared_ids = shared expert ids (broadcasted)
22     ids = concat(select_ids, shared_ids)
23     dt = gather dt using ids
24     spectral_bias = apply gamma and pad
25     dt = dt + dt_bias + spectral_bias
26     moe_loss = cv_squared(h)
27
28     # 5. Combine features again
29     zxbcdt = concat(zxbc, dt)
30
31     out = mamba_ssm_kernel(
32         input = zxbcdt,
33         weights = conv1d weights,
34         A, D, dt, etc.,
35         norm and activation config
36     )
37     out = reshape to (B, L, H, P)
38     diversity_loss_val = diversity_loss(out)
```

```
39    out = reshape to (B, L, D)
40    out = out_proj(out)
41
42    diversity_loss_val = diversity_loss(reshape y to (B, L, H, P))
43    out = out_proj(y)
44
45    return out, (moe_loss, diversity_loss_val)
```

Listing 1: Forward function in Prefill stage

```
1  def step(hidden_states, conv_state, ssm_state, cumsum_state, t_pos):
2      assert only one token at a time (sequence length == 1)
3      u = squeeze hidden_states
4
5      # 1. Input projection
6      zxbcdt = in_proj(u)
7      split zxbcdt into z0, x0, z, xBC, dt
8
9      # 2. Spectral residual routing
10     spectral_residual = u - (cumsum_state / (t_pos - 1))
11     udt = concat(spectral_residual, dt)
12     hb = h_proj(udt)
13     h, spectral_bias = split(hb)
14
15     # 3. MoE top-k selection
16     select_ids = topk(h)
17     shared_ids = fixed shared expert ids
18     ids = concat(select_ids, shared_ids)
19     dt = gather dt using ids
20     spectral_bias = gamma * concat(spectral_bias, zeros)
21     dt = dt + spectral_bias
22
23     # 4. Update cumsum state for next spectral residual
24     cumsum_state += u
25
26     # 5. Conv1D Step
27     xBC = causal_conv1d_update(xBC, conv_state, conv1d weights and bias)
28     split xBC into x, B, C
29
30     # 6. State-Space Model Step
31     # We use Expanded version for group/state-aware update
32     repeat/reshape A, dt, dt_bias, D, B, C
33     x = reshape x to (batch, heads, head_dim)
34     y = selective_state_update(ssm_state, x, dt, A, B, C, D, z, dt_bias)
35     y = reshape y to (batch, total_dim)
36
37     # 7. Output projection
38     out = out_proj(y)
39     return out.unsqueeze(1), conv_state, ssm_state, cumsum_state
```

Listing 2: Forward function in Decode stage

## B.2 COMPARISON TO OTHER FIELDS

**Comparison with MoE.** While our proposed method draws partial inspiration from the MoE framework, its core contribution lies in the filter bank interpretation from a GSP perspective. Specifically, our model interprets each head within a single architecture as a distinct filter and adaptively selects among them, making it most analogous to MoA (Zhang et al., 2022) among existing related works. In contrast, conventional MoE approaches route tokens across multiple separate architectures, leading to significantly larger model capacity and computational overhead. In this sense, our approach focuses on efficient utilization within a single architecture, whereas MoE methods entail learning and managing multiple parallel networks.

Regarding the loss function, we were inspired by the load balancing losses commonly used in MoE settings (Jin et al., 2024; Zhang et al., 2022; Dai et al., 2024). To ensure balanced selection, we apply loss terms both before and after routing, encouraging equitable utilization of filters.

From an interpretability standpoint, our GSP-based filter view allows the model's internal mechanisms to be understood more clearly as adaptive filtering. While implicit, this behavior manifests in observable differences in filtering effects across tokens and tasks (see Fig. 5 ), demonstrating the effectiveness of our formulation.

**Comparison with Adaptive Filtering.**   To better illustrate the effectiveness of our method, we present discussion comparing adaptive filtering approaches to *HADES*. Affirm (Wu et al., 2025) and our method differ mainly in their filtering strategies: Affirm uses explicit filtering by applying FFT-based domain transformation, enabling frequency-domain operations. In contrast, our approach leverages an implicit filtering mechanism, interpreting heads as filters from a GSP perspective. Without requiring domain transformation, our model adaptively selects token-specific filters, yielding performance improvements through flexible and interpretable routing. Focus (Lutati et al., 2023) takes a DSP-inspired approach by modeling SSMs as Infinite Impulse Response filters, transitioning from a Linear Time-Invariant to a Time-Variant perspective. It enables adaptive filtering through a modified STFT (chunked-FFT) and a hypernetwork that generates filters dynamically. While both Focus and our method introduce adaptivity via routing, the key difference is that Focus generates filters, whereas we select from pre-defined heads interpreted as filters. Moreover, Focus applies explicit frequency-domain filtering, while our method remains implicit, operating entirely in the original domain.

## C   EXPERIMENTAL SETUP

### C.1   TRAINING DETAILS

We adopt all baseline implementations from `flash-linear-attention` (Yang & Zhang, 2024). For fair comparison, all models are trained under identical conditions with 370M parameters exculding readout head on 200B tokens from the Pile dataset (Gao et al., 2020). Starting from 370M configuration, *HADES* yields smaller parameter count 218M. We use the AdamW optimizer with a peak learning rate of 48e-4, weight decay of 0.1, $\beta \in [0.9, 0.95]$ following Mamba2, and gradient clipping of 1.0. The learning rate follows a cosine annealing schedule with a warm-up phase of 375M tokens and a batch size of $2^{22}$ tokens (# sequences × sequence length) and the number of training steps as 47,042 (# tokens / # tokens in one batch) steps. All models employ the GPT-NeoX tokenizer with a vocabulary size of 50,277. For sequence modeling, we set the training length to 2K tokens. Our experiments were conducted on a computing server equipped with an AMD EPYC 9654 CPU (2 sockets, 192 cores, 384 threads, 1.5–3.7 GHz, L3 cache 768 MiB) and four NVIDIA A100 80GB PCIe GPUs with CUDA version 12.4. For our model, we used hyperparameter set of $H = 16$, $S = 8$, $\lambda_1 = 1e-3$, $\lambda_2 = 1e-3$, $\gamma = 25e-2$.

### C.2   EVALUATION

**Language Modeling and zero-shot Commonsense Reasoning.**   Following prior works (Gu & Dao, 2023; Yang et al., 2024a), we evaluate our method against five baseline models across two evaluation categories: WikiText (Wiki.) perplexity and zero-shot commonsense reasoning tasks. The commonsense tasks include LAMBADA (LMB.; Paperno et al. (2016)), PIQA Bisk et al. (2019), HellaSwag (Hella.; Zellers et al. (2019)), WinoGrande (Wino.; Sakaguchi et al. (2019)), ARC-easy (ARC-e) and ARC-challenge (ARC-c) Clark et al. (2018), BoolQ Clark et al. (2019), and OpenbookQA (OBQA.; Mihaylov et al. (2018)).

We measure perplexity (ppl) on WikiText and LAMBADA, normalized accuracy(acc_n) on HellaSwag and ARC-challenge, and standard accuracy (acc) on the remaining tasks (as normalized accuracy provides higher scores for most models on these tasks). Avg. denotes the averaged result of the accuracies and normalized accuracies of eight tasks together. All evaluations are conducted using `lm-evaluation-harness` (Liang et al., 2023). We provide details of the evaluation tasks below.

- WikiText (Merity et al., 2017): A dataset consisting of high-quality, clean text extracted from Wikipedia articles, commonly used to evaluate language modeling tasks by measuring a model's ability to predict and generate coherent and fluent text.

- LAMBADA (Paperno et al., 2016): A text completion task that measures a model's ability to predict the final word of a passage, requiring comprehension of the context, commonsense reasoning, as well as the ability to generate text coherently.

- PIQA (Bisk et al., 2019): A physical commonsense reasoning task focused on selecting the most plausible solution to everyday scenarios.

- HellaSwag (Zellers et al., 2019): A multiple-choice task that evaluates a model's ability to select the most coherent continuation of a given situation based on commonsense and narrative reasoning.
- WinoGrande (Sakaguchi et al., 2019): An expanded version of the Winograd Schema Challenge: a pronoun resolution task designed to test commonsense reasoning by identifying which noun a pronoun refers to in a given sentence.
- OpenbookQA (Mihaylov et al., 2018): A multiple-choice question answering task designed to test a model's understanding of elementary-level science facts and its ability to apply this knowledge to novel scenarios requiring reasoning and inference.
- ARC-easy (Clark et al., 2018): A subset of the AI2 Reasoning Challenge focusing on questions that require basic scientific and commonsense knowledge.
- ARC-challenge (Clark et al., 2018): A more difficult subset of the AI2 Reasoning Challenge that tests advanced reasoning and deep understanding of scientific and commonsense knowledge.
- BoolQ (Clark et al., 2019): A yes/no question answering dataset with 15,942 examples, derived from Google search queries, paired with Wikipedia passages.

**Passkey Retrieval.** For the passkey retrieval task, we adopt the task formulation from Chen et al. (2024). The evaluation is conducted across context lengths from 1K to 16K, with the target digit hidden at depths of 0% to 100% with the gap of 10% of each of these sequences. Assuming that each correct retrieval receives a score of 1 and each incorrect retrieval receives a score of 0, we compute the retrieval score as count out of 10, across all the depths overall context lengths. We did not apply any fine-tuning with longer sequences. We structure the prompt for the passkey retrieval task into four distinct components: task description, passkey, query, and dummy text.

- Task Description: This section defines the task for the model, instructing it to identify and memorize specific important information within a large amount of irrelevant text.

```
There is an important piece of information hidden inside a lot
of irrelevant text. Find it and memorize it. I will quiz you
about this important information.
```

- Passkey: This section provides the critical information (the passkey) that the model is required to memorize and retrieve.

```
The pass key is 15921. Remember it. 15921 is the pass key.
```

- Query: This part contains a direct question prompting the model to recall the passkey it memorized.

```
What is the pass key? The pass key is
```

- Dummy Text: This section consists of irrelevant text that serves as a placeholder, repeated until the full prompt length reaches the designated sequence length.

```
The grass is green. The sky is blue. The sun is yellow. Here we
go. There and back again.
```

# D    DETAILED BENCHMARKS AND MORE EVALUATIONS

## D.1    FULL RESULT OF LANGUAGE MODELING AND ZERO-SHOT COMMONSENSE REASONING

In this subsection, we report the full result of language modeling and zero-shot commonsense reasoning with standard error. For metrics aggregated using the mean (accuracy and normalized accuracy), the standard error was calculated using the conventional formula $\text{Standard Error} = \frac{s}{\sqrt{n}}$,

where s denotes the sample standard deviation, and n is the sample size. In contrast, for Perplexity, due to its potentially non-normal distribution, we employed a bootstrap method with 100 resampling iterations to estimate standard error, calculating the standard deviation of the resampled values.

Table 3: Performance comparison on language modeling and zero-shot common-sense reasoning with standard error (values in $(\cdot)$). The standard error values are are rounded to three decimal places. The best results are highlighted in **bold**, while the second-best results are underlined. Avg. denotes the result of accuracies and normalized accuracies over 8 tasks.

| Model | Wiki. ppl ↓ | LMB. ppl ↓ | LMB. acc ↑ | BoolQ acc ↑ | Hella. acc_n ↑ | Wino. acc ↑ | ARC-e acc ↑ | ARC-c acc ↑ | PIQA acc ↑ | OBQA. acc_n ↑ | Avg. 8 tasks ↑ |
|---|---|---|---|---|---|---|---|---|---|---|---|
| Linear Transformer | 45.43 (N/A) | 73.93 (4.168) | 24.06 (0.006) | 61.50 (0.009) | 28.20 (0.005) | 51.30 (0.014) | 42.05 (0.010) | 21.76 (0.012) | 60.55 (0.012) | 27.60 (0.020) | 39.63 (0.011) |
| RetNet | 34.12 (N/A) | 29.46 (1.046) | 35.36 (0.007) | 55.57 (0.009) | 31.31 (0.005) | 51.70 (0.014) | 44.49 (0.010) | 23.46 (0.012) | 62.40 (0.011) | 28.00 (0.020) | 41.54 (0.011) |
| DeltaNet | 33.25 (N/A) | 26.82 (0.908) | 35.75 (0.007) | 54.07 (0.009) | 31.40 (0.005) | 49.96 (0.014) | 44.11 (0.010) | 22.18 (0.012) | 63.60 (0.011) | **29.60** (0.020) | 41.33 (0.011) |
| Mamba1 | 47.51 (N/A) | 85.53 (3.321) | 22.43 (0.006) | **62.17** (0.009) | 28.71 (0.005) | 50.67 (0.014) | 42.09 (0.010) | 22.35 (0.012) | 60.72 (0.011) | 26.60 (0.020) | 39.47 (0.011) |
| Mamba2 | **31.34** (N/A) | 24.38 (0.820) | 36.46 (0.007) | 53.88 (0.009) | 32.62 (0.005) | 50.83 (0.014) | **45.29** (0.010) | **24.15** (0.013) | 63.44 (0.011) | 26.40 (0.020) | 41.63 (0.011) |
| *HADES* (Ours) | 31.48 (N/A) | **21.74** (0.727) | **39.24** (0.007) | 58.84 (0.009) | **32.82** (0.005) | **52.64** (0.014) | 45.03 (0.010) | 22.01 (0.012) | **63.93** (0.011) | 28.80 (0.020) | **42.91** (0.011) |

## D.2 MORE EXPERIMENTS

**Larger scale experiment.**  For generality, we conduct bigger scale experiment of our model in Table 4. To ensure a fair comparison, all models are trained under the same setup: 1.3B parameters and 30B tokens drawn from the FineWeb-Edu dataset Penedo et al. (2024). We adopt the AdamW optimizer with a peak learning rate of 4e-4, weight decay of 0.1, and apply gradient clipping at 1.0. The learning rate schedule uses cosine annealing with a warm-up phase of 1B tokens, and the batch size is fixed at 0.5M tokens. All models are trained with the Llama2 tokenizer, which has a vocabulary size of 32,000. For sequence modeling, the training sequence length is set to 4K tokens. *HADES* shows strong performance against baseline models with only 71.4% of parameters.

Table 4: Performance comparison on language modeling and zero-shot common-sense reasoning. The best results are highlighted in **bold**, while the second-best results are underlined. Avg. denotes the average of accuracies and normalized accuracies over 8 tasks. With only 71.4% of parameters compared to baseline models, *HADES* achieves comparable or even better performance.

| Model | Wiki. ppl ↓ | LMB. ppl ↓ | LMB. acc ↑ | PIQA acc ↑ | Hella. acc_n ↑ | Wino. acc ↑ | ARC-e acc ↑ | ARC-c acc ↑ | BoolQ acc ↑ | OBQA. acc_n ↑ | Avg. 8 tasks ↑ |
|---|---|---|---|---|---|---|---|---|---|---|---|
| RetNet (1.3B) | 22.45 | 21.84 | 38.70 | 69.04 | 47.73 | 52.72 | 63.68 | 33.36 | 60.61 | 36.60 | 50.31 |
| Mamba2 (1.3B) | **19.47** | 17.40 | 40.68 | 70.29 | **53.24** | 56.04 | **69.87** | **36.35** | 55.81 | 37.40 | 52.46 |
| DeltaNet (1.3B) | 19.77 | **16.64** | **41.78** | 70.95 | 51.09 | 54.70 | 67.63 | 34.47 | **61.19** | 38.40 | 52.53 |
| *HADES* (1B) | 20.41 | 17.22 | 41.18 | **71.33** | 51.85 | **56.35** | 68.48 | 34.81 | 60.73 | **38.60** | **52.92** |

**Comparison to mixture variant.**  We additionally compare our method against a mixture variant, MoM (Du et al., 2025), the mixture-of-experts extension of Gated DeltaNet (Yang et al., 2025), itself an adaptation of Mamba2. For fairness, we use the same 370M configuration ($d_{\text{model}} = 1024, n_{\text{layer}} = 24$), and we follow the official configuration of MoM, which employs 4 experts. All other training and evaluation settings follow our original setup in Appendix C. Across all benchmarks, in Table 5, *HADES* consistently achieves comparable or higher average performance compared to MoM, demonstrating the effectiveness of our approach even relative to mixture-based variants.

**Robustness over seed sweep.**  To further assess robustness of *HADES*, we conducted an additional seed sweep. Specifically, we evaluated *HADES* and the primary baseline, Mamba2, across three random seeds. All other training and evaluation settings follow our original setup in Appendix C. The results are summarized in Table 6, where our model consistently maintains strong performance with low variance across seeds, demonstrating robustness to initialization.

Table 5: Performance comparison on language modeling and zero-shot common-sense reasoning. The best results are highlighted in **bold**, while the second-best results are underlined. Avg. denotes the average of accuracies and normalized accuracies over 8 tasks.

| Model | Wiki. ppl ↓ | LMB. ppl ↓ | LMB. acc ↑ | PIQA acc ↑ | Hella. acc_n ↑ | Wino. acc ↑ | ARC-e acc ↑ | ARC-c acc ↑ | BoolQ acc ↑ | OBQA. acc_n ↑ | Avg. 8 tasks ↑ |
|---|---|---|---|---|---|---|---|---|---|---|---|
| MoM | 31.58 | 23.28 | **40.40** | 62.57 | **32.99** | **52.64** | 44.91 | **23.89** | 52.78 | 27.20 | 42.17 |
| *HADES* | **31.48** | **21.74** | 39.24 | **63.93** | 32.82 | **52.64** | **45.03** | 22.01 | **58.84** | **28.80** | **42.91** |

Table 6: Performance comparison on language modeling and zero-shot common-sense reasoning with mean and standard error over three seeds (values in ( · )). The standard error values are are rounded to three decimal places. The best results are highlighted in **bold**, while the second-best results are underlined. Avg. denotes the result of accuracies and normalized accuracies over 8 tasks.

| Model | Wiki. ppl ↓ | LMB. ppl ↓ | LMB. acc ↑ | BoolQ acc ↑ | Hella. acc_n ↑ | Wino. acc ↑ | ARC-e acc ↑ | ARC-c acc ↑ | PIQA acc ↑ | OBQA. acc_n ↑ | Avg. 8 tasks ↑ |
|---|---|---|---|---|---|---|---|---|---|---|---|
| Mamba2 | **30.64** (0.752) | **22.57** (1.662) | **37.85** (0.014) | 54.63 (0.007) | **33.27** (0.007) | 51.15 (0.006) | **45.34** (0.004) | **23.29** (0.009) | 63.04 (0.014) | **27.93** (0.014) | 42.06 (0.004) |
| *HADES* | 33.41 (1.721) | 25.57 (3.482) | 37.10 (0.022) | **59.67** (0.009) | 31.70 (0.010) | **51.85** (0.011) | 44.84 (0.008) | 22.67 (0.006) | **63.15** (0.007) | 27.87 (0.013) | **42.37** (0.005) |

# E EXTENDED MODEL ANALYSES

## E.1 ABLATION STUDIES

In this subsection, we report the full result of ablation studies. We test variations of our model with same training and evaluation setting in Appendix C. In Table 7, our ablation studies demonstrate both the robustness and tunability of our model.

Table 7: Full result for ablation studies. Avg. denotes the averaged result of the accuracies and normalized accuracies of eight tasks together.

| Methods | Wiki. ppl ↓ | LMB. ppl ↓ | LMB. acc ↑ | PIQA acc ↑ | Hella. acc_n ↑ | Wino. acc ↑ | ARC-e acc ↑ | ARC-c acc ↑ | BoolQ acc ↑ | OBQA. acc_n ↑ | Avg. 8 tasks ↑ |
|---|---|---|---|---|---|---|---|---|---|---|---|
| *HADES* (Ours) | **31.48** | **21.74** | **39.24** | **63.93** | 32.82 | 52.64 | **45.03** | 22.01 | 58.84 | 28.80 | **42.91** |
| w/o $\mathcal{L}_{\text{balance}}$ | 34.73 | 26.84 | 36.77 | 62.68 | 30.96 | 50.75 | 43.22 | **22.70** | 59.27 | 26.20 | 41.57 |
| w/o $\mathcal{L}_{\text{diversity}}$ | 33.83 | 27.40 | 36.04 | 62.46 | 31.48 | 51.38 | 44.87 | 22.61 | 59.94 | 28.40 | 42.15 |
| Only Shared Filters | 34.55 | 27.64 | 35.75 | 62.40 | 31.39 | **52.88** | 44.23 | 24.23 | **60.83** | 26.00 | 42.21 |
| Only Expert Filters | 36.34 | 30.12 | 34.89 | 61.53 | 30.30 | 52.41 | 44.49 | **22.70** | 58.29 | **28.80** | 41.68 |
| 25 % Shared | 35.24 | 29.53 | 34.64 | 62.79 | 30.89 | 50.43 | 43.35 | 23.81 | 59.48 | 28.20 | 41.70 |
| 75 % Shared | 35.92 | 31.67 | 34.08 | 62.35 | 30.12 | 50.83 | 42.89 | 23.21 | 61.80 | 28.40 | 41.71 |
| Fixed | 34.55 | 27.64 | 35.75 | 62.40 | 31.39 | **52.88** | 44.23 | **24.23** | **60.83** | 26.00 | 42.21 |
| Random Routing | 35.78 | 32.77 | 33.17 | 61.97 | 30.47 | 52.49 | 43.31 | 23.12 | 55.72 | 28.00 | 41.03 |
| Input-only | 34.17 | 23.95 | 38.40 | 63.71 | 31.60 | 51.22 | 43.56 | 23.04 | 58.35 | 27.80 | 42.21 |
| Gumbel Softmax Top-K | 34.83 | 27.21 | 36.95 | 62.19 | 31.39 | 50.51 | 43.43 | 22.35 | 58.07 | 28.20 | 41.64 |
| Weighted aggregation (MoH) | 36.73 | 32.03 | 34.95 | 61.75 | 30.19 | 50.75 | 44.36 | 22.53 | 60.92 | 28.00 | 41.68 |
| Position Bias | 30.23 | 21.93 | 38.50 | **63.93** | **33.08** | 51.70 | 43.73 | **22.27** | 53.39 | **30.60** | 42.15 |
| No Bias | 34.57 | 28.79 | 34.91 | 63.38 | 31.24 | 50.67 | 42.68 | 22.35 | 56.85 | 26.80 | 41.11 |

**On Auxiliary Losses and Filter Configurations.** Interestingly, without $\mathcal{L}_{\text{balance}}$, performance drops as filter selection becomes overly concentrated on a few filters, leaving the rarely selected filters under-trained and preventing them to learn dynamics effectively when they are eventually chosen. We also evaluate the impact of different filter configurations: Using Only Shared Filters and Using Only Expert Filters. Using only shared filters outperforms using only expert filters, as shared filters consistently capture global low-frequency information, while expert filters adaptively capturing low and high frequency information.

**On Filter selection and Delta modulation.** We tried more ablation on filter selection and delta modulation. For filter selection, "Top-Q" refers to the routing mechanism used in our proposed method, *HADES*, where the top-ranked filters are adaptively selected. "Fixed" denotes the setup with no routing—i.e., a fixed set of filters is always used regardless of input. "Random" indicates that filters are selected at random without regard to token-specific information. We additionally performed "Input-only", "Gumbel Softmax Top-K" and "Weighted Aggregation". "Input-only" refers to the routing mechanism which only use input sequence to get top-ranked filters. "Gumbel Softmax Top-K" is where Top-K filters are selected with Top-K selection itself is being trained. "Weighted Aggregation" means output is aggregated via linear projection instead of simple aggregation, which can be interpreted as variant of MoH (Jin et al., 2024).

For delta modulation, "Spectral Bias" refers to the biasing scheme originally used in *HADES*, which modulates $\Delta$ based on learned spectral residual. "Position Bias" incorporates positional information of each token into the bias term, enabling location-aware modulation. "No Bias" denotes the variant where no additional modulation is applied to $\Delta$.

On filter selection, we observed that average performance of "Fixed" was better than that of purely random selection. Also, "Input-only" showed reasonable performance with simple selection. However, leveraging the token-level select score to guide the selection (*HADES*) yielded the strongest results, as it allowed the model to adapt to the specific characteristics of the input. Regarding delta modulation, introducing a positional bias was more beneficial than using no bias at all. Yet, instead of relying solely on absolute positions, incorporating delta-based information—thereby reflecting the relative importance of the input—proved to be more effective in achieving superior performance.

### E.2 SENSITIVITY STUDIES

Table 8: Sensitivity Studies. Avg. denotes the averaged result of the accuracies and normalized accuracies of eight tasks together.

| Hyper param. | | Wiki. ppl↓ | LMB. ppl↓ | LMB. acc↑ | PIQA acc↑ | Hella. acc_n↑ | Wino. acc↑ | ARC-e acc↑ | ARC-c acc↑ | BoolQ acc↑ | OBQA. acc_n↑ | Avg. 8 tasks↑ |
|---|---|---|---|---|---|---|---|---|---|---|---|---|
| | 8 | 39.52 | 37.58 | 32.93 | 60.88 | 29.63 | 50.28 | 42.00 | 22.27 | 61.10 | 24.60 | 40.46 |
| $H$ | 16 | 31.48 | 21.74 | 39.24 | 63.93 | 32.82 | 52.64 | 45.03 | 22.01 | 58.84 | 28.80 | 42.91 |
| | 24 | 33.11 | 26.73 | 35.40 | 62.57 | 31.99 | 50.75 | 44.36 | 23.29 | 48.93 | 28.60 | 40.74 |
| | 0.15 | 34.81 | 29.07 | 34.78 | 61.86 | 31.33 | 51.22 | 43.64 | 23.04 | 60.34 | 26.60 | 41.60 |
| $\gamma$ | 0.25 | 31.48 | 21.74 | 39.24 | 63.93 | 32.82 | 52.64 | 45.03 | 22.01 | 58.84 | 28.80 | 42.91 |
| | 0.35 | 33.96 | 28.42 | 35.84 | 63.06 | 31.69 | 52.33 | 43.52 | 22.87 | 50.09 | 28.20 | 40.95 |

We conduct a sensitivity analysis to assess our model's robustness to hyperparameters. First, We train our model varing $\gamma \in [0.15, 0.35]$ in increments of 0.1 while other settings are fixed (See Appendix C.1). We use same evaluation settings in Appendix C.2. Table 8 show that performance on language modeling and zero-shot commonsense reasoning benchmarks remains stable, even as higher $\gamma$ increases bias influence, demonstrating model robustness. We then examine the sensitivity of hyperparameter $H$ by varying the number of active filters among the total of 32. We test $H \in \{8, 16, 24\}$ while keeping other settings fixed. The best performance is achieved with $H = 16$, followed by 24 and 8, supporting our hypothesis that an optimal number of filters enhances information flow. As shown in Table 8, even with a drastically reduced model size of approximately 38.64% (143M) in the $H = 8$ setting, our model maintains performance comparable to the optimal hyperparameter configuration and even outperforms it on two tasks. It is worth noting that more filters does not mean better performance: $H = 24$ failed to outperform both $H = 8$, $H = 16$ setting. This result highlights that selective filter activation can effectively reduce redundancy without sacrificing performance, demonstrating the efficiency of our filter bank approach.

### E.3 CKA ANALYSIS ON MAMBA2 AND *HADES*

We additionally analyze the organization of dynamic filtering behaviors using linear centered kernel alignment (CKA) (Kornblith et al., 2019) on both Mamba2 and *HADES*. Given two filter outputs $X, Y \in \mathbb{R}^{n \times d}$, where each row corresponds to a sequence position and each column to a feature

dimension, we first mean-center the features: $\bar{X} = X - \mathbf{1}X_{\text{mean}}$, $\bar{Y} = Y - \mathbf{1}Y_{\text{mean}}$. The linear CKA between the two filter outputs is computed as:

$$\text{CKA}(X, Y) = \frac{\|\bar{X}^\top \bar{Y}\|_F^2}{\|\bar{X}^\top \bar{X}\|_F \ \|\bar{Y}^\top \bar{Y}\|_F}. \tag{32}$$

For each layer, we compute CKA across all filter output pairs, and use the mean off-diagonal CKA as a single redundancy score.

As shown in Fig. 8(d), the CKA heatmap of Mamba2 reveals that Mamba2 contains a substantial number of redundant filters, with many filter pairs exhibiting high similarity. This redundancy indicates that, although Mamba2 possesses a dynamic filtering mechanism, a large portion of its filters operate in overlapping regions and fail to specialize effectively. In contrast, *HADES* shows a disappearance of these repetitive structures, suggesting that our model avoids redundant filters and instead selects the filters that are genuinely needed. When comparing the overall similarity distributions, *HADES* is noticeably skewed toward lower similarity values, further demonstrating that it achieves a more diverse and well-differentiated filter selection.

### E.4 MORE VISUALIZATION ON SPECTRUM OF INPUT AND OUTPUT SEQUENCES

**Details.** To validate the filter behaviors, we analyze the spectrum of the input and output sequences processed by *HADES* and Mamba2. We compute the sequence spectrum for Fig. 5, 7(a), 7(b), 9, and 12 in the following way. Given a sequence $x$, we obtain its frequency representation $\tilde{x} = \mathcal{F}x$, where $\mathcal{F}$ denotes the 1D discrete Fourier transform applied along the temporal dimension. We then measure the amplitude of each frequency component to quantify how the input and output sequences differ in their spectral distribution. For ease of comparison, each spectrum is normalized by its maximum amplitude (max scaling). This enables us to assess how the shared and expert filters modify the frequency content of the processed sequences.

**More Visualization.** In Fig. 9, we provide additional spectrum visualizations of the input and output sequences processed by *HADES* across various layers. The input sequence is taken from a randomly sampled sentence from the Pile dataset.

### E.5 MORE VISUALIZATIONS ON THE FREQUENCY RESPONSE OF FILTER

**Details.** To analyze the frequency behavior of the filter itself, we compute the frequency response of the *HADES* filter and compare it with that of Mamba2. The frequency responses shown in Fig. 1, Fig. 6, and Fig. 10 are computed as follows. Following the formulation in Mamba2 (Dao & Gu, 2024), both Mamba2 and *HADES* can be interpreted as linear sequence-to-sequence operators whose entire computation is equivalently captured by a transformation matrix $M$. This matrix plays the same role as the attention matrix in Transformers, serving as the full linear operator acting on the sequence, and therefore its frequency response can be analyzed in exactly the same manner as that of an attention matrix (Wang et al., 2022). Given a filter represented as a transformation matrix $M$, we characterize its frequency behavior through its Fourier-domain representation $\Lambda = \mathcal{F}M\mathcal{F}^{-1}$, where $\mathcal{F}$ denotes the discrete Fourier transform operator. The magnitude of each frequency response is evaluated using the norm $\|\Lambda_i\|_2$, allowing us to assess how strongly the filter amplifies or suppresses each frequency component. For fair comparison across filters, the spectra of the processed sequences are additionally $\|\cdot\|_2$-normalized, ensuring that differences arise from the filters themselves rather than scale variations in the underlying sequences.

**More Visualization.** We provide additional filter frequency response visualizations in Fig. 10, including responses of Mamba2 and *HADES* from multiple layers. Mamba2's learned filters remain low-pass and highly redundant across layers, collapsing into nearly identical spectral kernels. *HADES* introduces rippled high-frequency responses and broader spectral variability, enabling the model to capture finer structure and richer local detail.

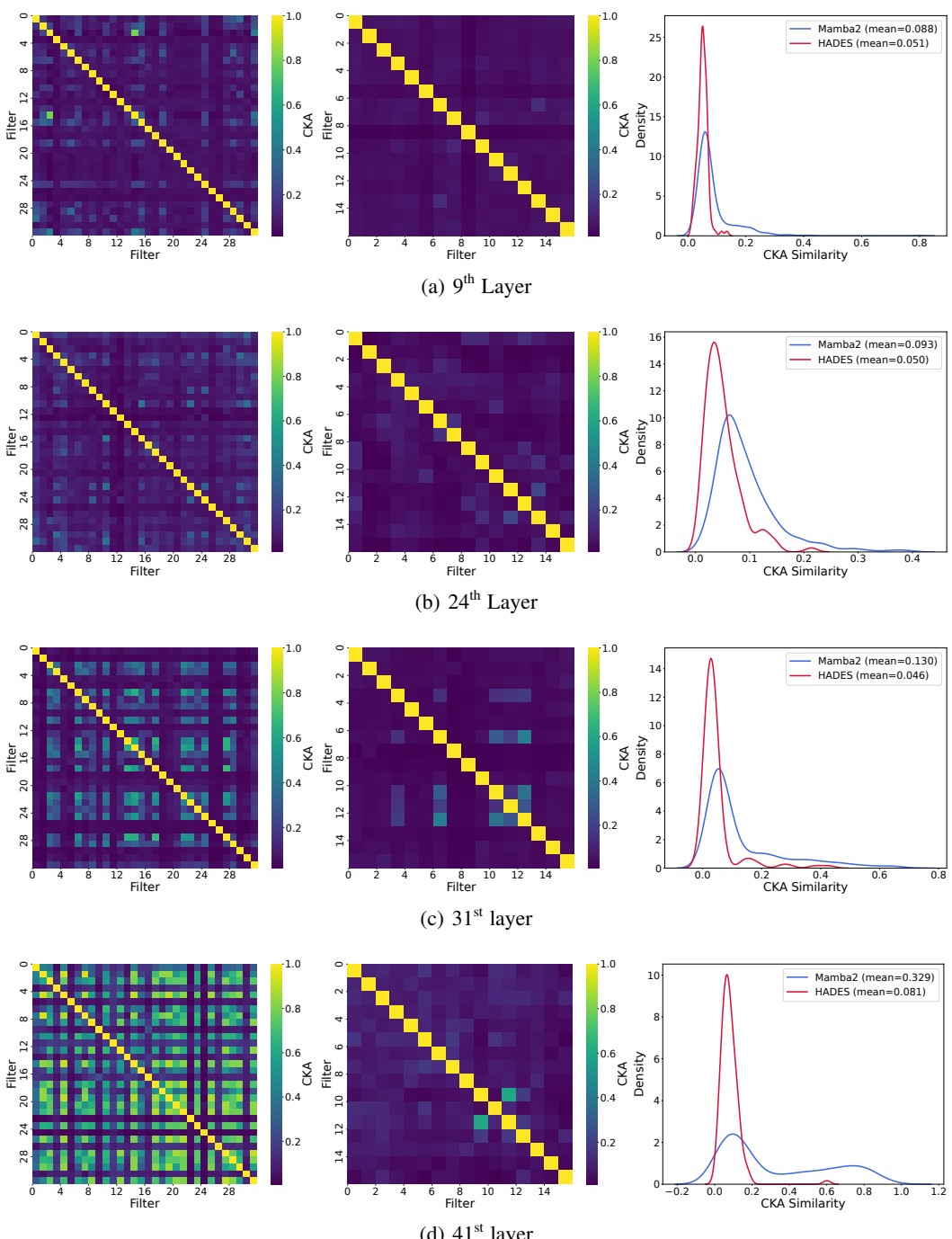

Figure 8: CKA analysis on filter outputs of Mamba2 and *HADES*. Left: Mamba2. Center: *HADES*. Right: comparison of distribution.

# F EFFICIENCY AND COMPUTATIONAL COMPLEXITY

## F.1 ANALYSIS ON COMPUTATIONAL TIME AND MEMORY USAGE

We evaluate the efficiency of our method in terms of computation inference time and memory usage. Specifically, we measure the inference time and memory consumption at every 10 steps for 100 steps,

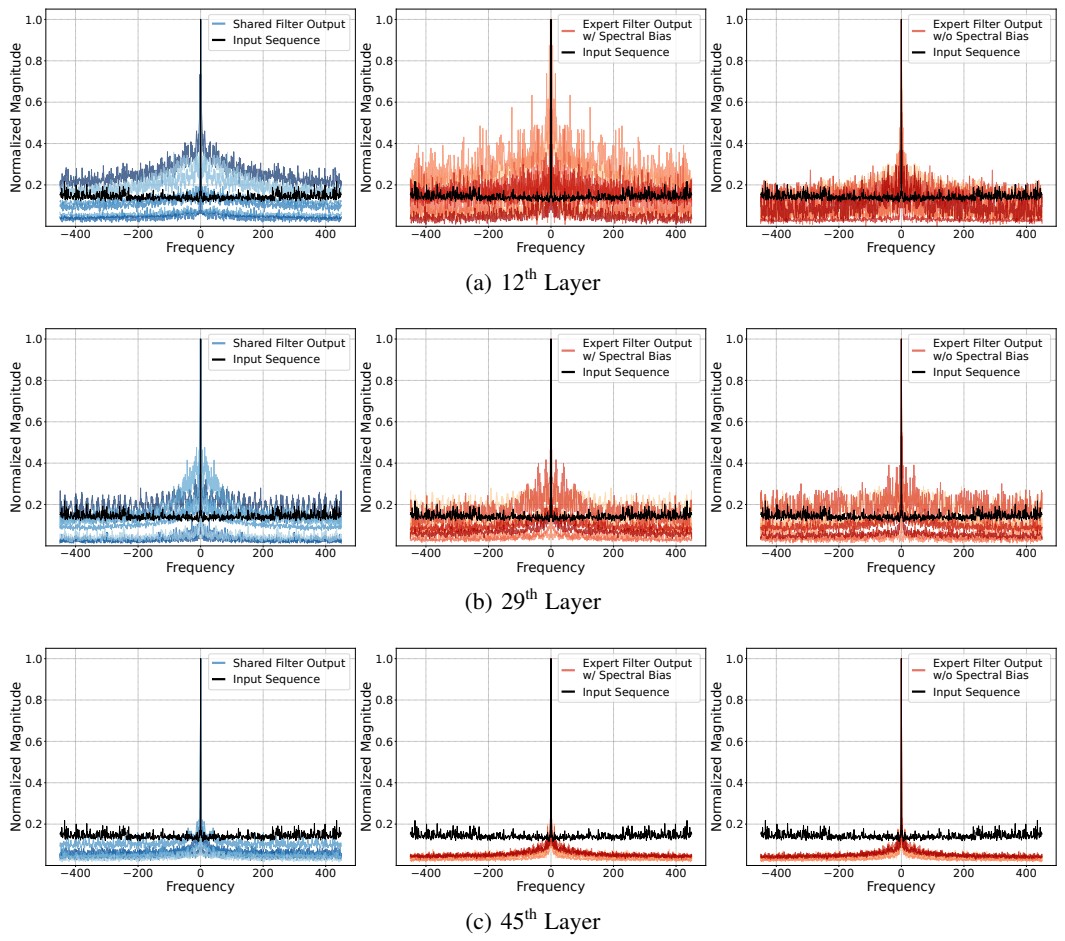

Figure 9: Frequency spectrum analysis of filter outputs. Left: Shared filter outputs. Center: Expert filters with spectral bias outputs. Right: Expert filters without spectral bias outputs.

Table 9: Inference Time and Memory Usage Comparison.

| Model
# Params. | Linear Transformer
370M | RetNet
370M | DeltaNet
370M | Mamba
370M | Mamba2
370M | *HADES*
218M |
|---|---|---|---|---|---|---|
| Inference Time (sec) | 2.19 | 2.86 | 2.57 | 4.73 | 3.43 | 2.49 |
| Memory (GB) | 5.41 | 6.17 | 5.59 | 5.56 | 6.20 | 4.50 |

using a sequence length of 1024 and a batch size of 32, and report the average values in Table 9. In this scenario, *HADES*(218M) demonstrates a 1.37x speed improvement and 1.37x lower memory usage compared to Mamba2. Furthermore, when compared to other baselines, our approach not only achieves faster processing speeds but also significantly reduces memory consumption, highlighting its efficiency. Additionally, we also examined our model matching the 370M parameter configuration for a direct comparison (Time: 3.45s, Memory: 5.91GB). However, this setting requires using 90 layers, about 1.8 times larger number relative to Mamba2. We argue that such a configuration is less relevant to the practical setting where *HADES* is used as a parameter-efficient drop-in replacement for a given 370M model. Therefore, our primary evaluation focuses on configurations with matched hidden dimensions and an equal number of layers.

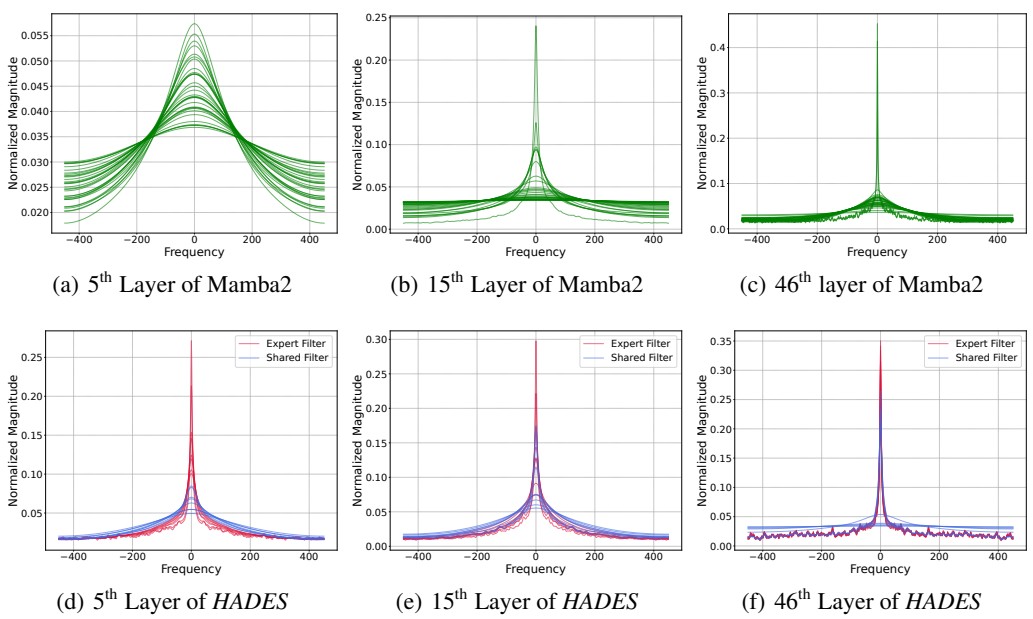

Figure 10: Frequency responses of Mamba2 and *HADES*.

## F.2 LATENCY ANALYSIS

For more analysis on different model size and sequence length, we recorded the processing time for both prefill and decode stage and reported the average over 21 runs. We report results in Table 10 Also, we evaluated routing overhead in the prefill stage of sequence length 2048, averaged over 100 steps of forward operation in Table 11.

Table 10: Average prefill and decode time under various configuration settings. Latency is reported in seconds. We utilize half of the selected filters as shared filters and the remaining half as expert filters. The selection ratio denotes the percentage of filters chosen out of the total available filters.

| Model | Setting | Selection Ratio | | |
|---|---|---|---|---|
| | | 50% | 25% | 75% |
| **370M** | Prefill (1K) | 0.3997 | 0.3374 | 0.4969 |
| | Prefill (2K) | 0.5936 | 0.4572 | 0.7869 |
| | Prefill (3K) | 0.6043 | 0.4765 | 0.8033 |
| | Decode | 0.0004 | 0.0003 | 0.0004 |
| **1.3B** | Prefill (1K) | 0.9981 | 0.5165 | 0.7742 |
| | Prefill (2K) | 1.4274 | 1.9154 | 1.9185 |
| | Prefill (3K) | 1.4337 | 1.9697 | 1.9614 |
| | Decode | 0.0067 | 0.0021 | 0.0045 |

Table 11: Latency overhead of running routers (seqlen 2048, averaged over 100 step)

| Metric | Value |
|---|---|
| Avg Routing Time (ms) | 0.0071 |
| Avg Total Time (ms) | 0.2900 |
| Avg Routing Time Ratio (%) | 2.4371% |

### F.3 THEORETICAL COMPUTATIONAL COMPLEXITY ANALYSIS

For a comprehensive analysis of overall efficiency, we calculated the computational complexity of our method in Table 12. Our architecture introduces additional operations for filter selection and Delta modulation, but otherwise performs the same computations as the original Mamba2. Crucially, unlike the original model that utilizes the entire set of filters, our approach employs a reduced number of filters, resulting in a corresponding reduction in computational cost (i.e. $H << M$). Since the total complexity mainly depends on the hidden dimension $d$, the introduced filter selection and Delta modulation computations incur only minimal overhead relative to the savings, preserving the overall efficiency of the model. Here, $T$ denotes the input sequence length, $d$ the hidden dimension, $M$ the total number of filters, $P$ the filter dimension, $H$ the number of selected filters, $S$ the number of shared filters, $E$ the number of expert filters, $d_{\text{conv}}$ the convolution kernel dimension, $C_{\text{in}}$ the number of input channels, and $C_{\text{out}}$ the number of output channels. We also analyze step-by-step breakdown of Filter selection and Delta modulation operation.

Table 12: Complexity comparison for each operation in prefill and decode stages. Arrows ($\rightarrow$) indicate improved complexity.

| Operation | Prefill |
|---|---|
| In Projection | $O(Td^2) = O(TMPd) \rightarrow O(THPd)$ |
| 1D Convolution | $O(TC_{\text{in}}C_{\text{out}}d_{\text{conv}}) = O(T(MP+N)^2 d_{\text{conv}}) \rightarrow O(T(HP+N)^2 d_{\text{conv}})$ |
| SSM Kernel | $O(T \log T \cdot M) \rightarrow O(T \log T \cdot H)$ |
| Out Projection | $O(Td^2) = O(TMPd) \rightarrow O(THPd)$ |
| RMS Norm | $O(Td) = O(TMP) \rightarrow O(THP)$ |
| HADES Ops. | $O(T(d+M)(M+H-2S))$ |

| Operation | Decode |
|---|---|
| In Projection | $O(d^2) = O(MPd) \rightarrow O(HPd)$ |
| 1D Convolution | $O(C_{\text{in}}C_{\text{out}}d_{\text{conv}}) = O((MP+N)^2 d_{\text{conv}}) \rightarrow O((HP+N)^2 d_{\text{conv}})$ |
| SSM Kernel | $O(MN) \rightarrow O(HN)$ |
| Out Projection | $O(d^2) = O(MPd) \rightarrow O(HPd)$ |
| RMS Norm | $O(d) = O(MP) \rightarrow O(HP)$ |
| HADES Ops. | $O((d+M)(M+H-2S))$ |

| *HADES* Operation | Complexity |
|---|---|
| Residual calculation | $O(Td)$ |
| Projecting selection score | $O(T(d+M)(M+H-2S))$ |
| Top-Q selection | $O(TE \log E)$ |
| Spectral bias calculation | $O(TH)$ |
| Delta Modulation | $O(TH)$ |

### F.4 PARAMETER REDUCTION ANALYSIS

As you've previously mentioned, the majority of parameters in Mamba2 originate from the linear and convolution layers. In our approach, since parameters are instantiated only for the candidate filters selected at the time of filter count determination, we are able to construct a significantly lighter-weight model compared to the original. A detailed parameter breakdown is provided in Table 13. Here, we use the following notation: $T$ denotes the sequence length, $d$ the hidden dimension, $M$ the total number of filters, $P$ the filter dimension, $H$ the number of selected filters, $S$ the number of shared filters, and $d_{\text{conv}}$ the convolution dimension.

In case of 370M parameters, $d = 1024$, $M = 32$, $H = 16$, $P = 64$, $N = 128$, $d_{\text{conv}} = 4$, $n_{\text{layer}} = 48$. Therefore, resulting parameter size would be $368,346,624 - 150,407,424 = 217,939,200 \simeq 218M$.

### F.5 FLOPS ANALYSIS

We analyze the computational complexity of *HADES* by decomposing it into two primary components: the mixer complexity and the routing overhead. The mixer complexity follows the Mamba2 structure

Table 13: Parameter complexity of each component. Arrows ($\rightarrow$) indicate reduction from $M$ to $H$.

| Component | Parameters |
|---|---|
| in_proj linear | $d \cdot (2 \cdot 2d + 2N + M) = d \cdot (2MP + 2N + M) \;\rightarrow\; d \cdot (2HP + 2N + M)$ |
| conv1d | $(2d + N) \cdot d_{\text{conv}} = (MP + N) \cdot d_{\text{conv}} \;\rightarrow\; (HP + N) \cdot d_{\text{conv}}$ |
| out_proj | $2d \cdot d = MP \cdot d \;\rightarrow\; HP \cdot d$ |
| rms norm | $2d = MP \;\rightarrow\; HP$ |
| ssm params | $3M \;\rightarrow\; 3H$ |
| Added params in *HADES* | $(d + M) \cdot (M + H - 2S) + 2$ |

| Name | Mixer Parameters |
|---|---|
| Mamba2 | $M[P(3d + d_{\text{conv}} + 1) + d + 3] + N(2d + d_{\text{conv}})$ |
| HADES | $H[P(3d + d_{\text{conv}} + 1) + d + 3] + N(2d + d_{\text{conv}}) + (d + M)(M + H - 2S) + 2$ |
| Reduction | $(M - H)[P(3d + d_{\text{conv}} + 1) + d + 3] - (d + M)(M + H - 2S) - 2$ |

Table 14: Per-token FLOPs of Mamba2 and *HADES* (excluding *HADES* selection module).

| Operation | Mamba2 | HADES | $\Delta$ (Reduction) |
|---|---|---|---|
| In-projection | $2d(2MP + 2N + M)$ | $2d(2HP + 2N + M)$ | $4dP(M - H)$ |
| 1D Convolution | $2(MP + N)d_{\text{conv}}$ | $2(HP + N)d_{\text{conv}}$ | $2(M - H)Pd_{\text{conv}}$ |
| Out-projection | $2MPd$ | $2HPd$ | $2(M - H)Pd$ |
| RMS Norm | $c_{\text{rms}}MP$ | $c_{\text{rms}}HP$ | $c_{\text{rms}}P(M - H)$ |
| SSD (SSM Core) | $c_{\text{ssd}}MN \log N$ | $c_{\text{ssd}}HN \log N$ | $c_{\text{ssd}}(M - H)N \log N$ |

but operates on a reduced set of $H$ active filters ($H < M$), serving as the dominant cost factor. The routing overhead encompasses the lightweight operations introduced by the selection mechanism, such as score computation and delta modulation. The following sections detail each component and quantify the overall efficiency gain compared to Mamba2.

**Mixer Complexity.** The core Mamba2 mixer consists of several filter-wise components including in-projection, 1D convolution, out-projection, RMSNorm, and the SSM kernel. Each filter maintains its own parameters and state, so these computations are applied independently for every filter, making their cost linearly proportional to the filter count $M$. The FLOPs of the Mamba2 mixer are:

$$\text{FLOPs}_{\text{Mamba2}} = T \cdot \left( M \cdot \text{FLOPs}_{\text{filter}} + \text{FLOPs}_{\text{const}} \right)$$

$$\text{FLOPs}_{\text{filter}} = 2\left( P(3d + d_{\text{conv}} + 1) + d + 3 \right) + c_{\text{ssd}}N \log N$$

$$\text{FLOPs}_{\text{const}} = 2N(2d + d_{\text{conv}})$$

HADES uses only $H$ filters from the original $M$. Since all filter-dependent computations scale linearly with the number of filters, replacing $M$ with $H$ yields:

$$\text{FLOPs}_{\text{HADES-mixer}} = T \cdot \left( H \cdot \text{FLOPs}_{\text{filter}} + \text{FLOPs}_{\text{const}} \right)$$

**Routing Overhead.** Beyond the mixer, *HADES* performs several additional but lightweight per-token operations: residual computation, selection score projection, top-$Q$ selection, spectral bias, and delta modulation. Their individual FLOPs per token are summarized in Table 15. Since these terms are small and heterogeneous, we denote their total cost over the sequence length $T$ as $\text{FLOPs}_{\text{HADES-ops}}$ rather than collapsing them into a single closed-form expression.

**Complexity Comparison.** Combining the mixer and routing costs gives the total computational complexity:

$$\text{FLOPs}_{\text{HADES}} = \text{FLOPs}_{\text{HADES-mixer}} + \text{FLOPs}_{\text{HADES-ops}} \tag{33}$$

$$= T \cdot \left( H \cdot \text{FLOPs}_{\text{filter}} + \text{FLOPs}_{\text{const}} \right) + \text{FLOPs}_{\text{HADES-ops}}. \tag{34}$$

To quantify the computational benefit, we analyze the reduction in FLOPs:

$$\Delta_{\text{FLOPs}} = \text{FLOPs}_{\text{Mamba2}} - \text{FLOPs}_{\text{HADES}} \tag{35}$$

$$= \underbrace{T \cdot (M - H) \cdot \text{FLOPs}_{\text{filter}}}_{\text{Main Savings}} - \underbrace{\text{FLOPs}_{\text{HADES-ops}}}_{\text{Routing Overhead}} \tag{36}$$

Table 15: Additional per-token FLOPs introduced by the *HADES* selection mechanism.

| HADES Operation | FLOPs/token |
|---|---|
| Residual computation | $2d$ |
| Selection score projection | $2(d + M)(M + H - 2S)$ |
| Top-$Q$ selection | $c_{\text{top}} E \log E$ |
| Spectral bias | $2H$ |
| Delta modulation | $2H$ |

Since the mixer cost $\text{FLOPs}_{\text{filter}}$ involves heavy $O(N \log N)$ operations while the routing overhead $\text{FLOPs}_{\text{HADES-ops}}$ consists only of lightweight linear projections, the savings term dominates the overhead (Main Savings $\gg$ Routing Overhead). This guarantees a substantial net reduction in computational complexity. Consequently, by disregarding the negligible overhead, we can approximate the total FLOPs of *HADES* as scaling proportionally with the filter ratio:

$$\text{FLOPs}_{\text{HADES}} \approx \frac{H}{M} \cdot \text{FLOPs}_{\text{Mamba2}}. \tag{37}$$

This approximation succinctly captures the efficiency gain obtained by activating only $H$ out of $M$ filters.

### F.6  TRAINING COMPLEXITY ANALYSIS

To clearly illustrate the training dynamics of *HADES*, we present the loss convergence landscape and training time observed during training. The training was conducted under the experimental settings described in Appendix C. Thanks to its parameter reduction, in Fig. 11(b), *HADES* exhibits faster training speed compared to both Mamba2 and Mamba1, and the loss curve in Fig. 11(a) shows stable and natural convergence throughout training.

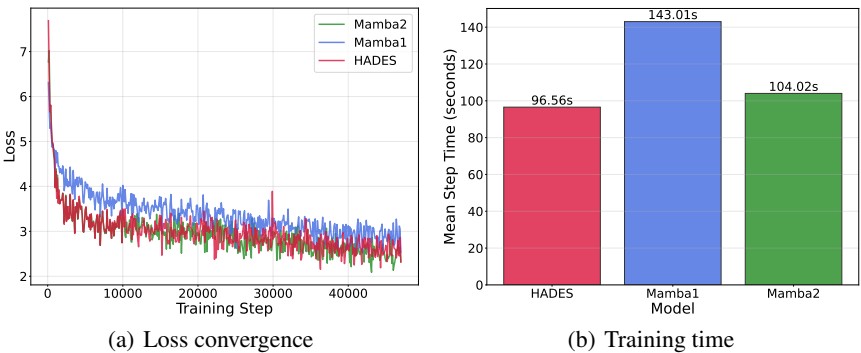

(a) Loss convergence          (b) Training time

Figure 11: Comparison of training behavior across models. Training sequence length was 2048, parameter size 370M, average time per 100 training steps for one epoch.

## G  LIMITATION AND FUTURE WORKS

While this study introduces a novel perspective on Mamba2 by reinterpreting it as a filter bank through the lens of GSP and proposes a new design methodology, there are some limitations. Although our design is inspired by GSP principles, we have not explicitly enforced spectral properties within the model. Instead, we adopt an implicit design approach, where spectral characteristics are indirectly encouraged with slight modification of biases. Explicitly enforcing spectral properties could lead to overly rigid behavior, which may hinder model performance. Our current approach aims to maintain flexibility while subtly guiding the model toward desirable spectral behavior. For future work, we aim to conduct a theoretical analysis of the advantages of explicit spectral design and explore new methods for biasing and filter selection that directly leverage these properties. Such an investigation could lead to more robust and interpretable state-space models.

# H   MORE VISUALIZATIONS

In this section, we provide more visualizations on our method. We compare output difference regarding $\gamma$-value in Fig. 12 following procedure in Appendix E.4. We applied Fourier transform to filter outputs obtained from a randomly sampled sentence from the Pile dataset. Although the spectrum of filter outputs varies with different values of $\gamma$, comparing the outputs with and without the bias consistently shows that the bias behaves as intended.

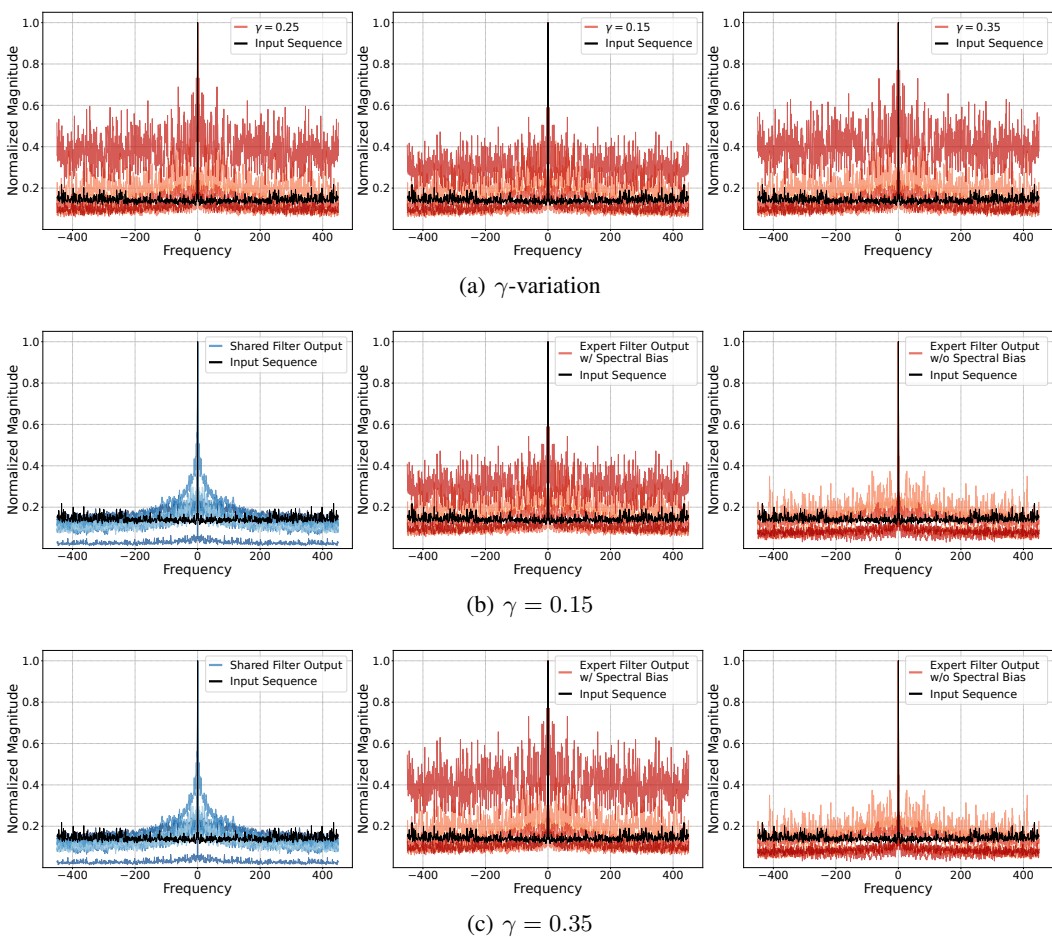

Figure 12: (a) $\gamma$-variation on expert filters with spectral bias outputs on 13[th] layer. (b) and (c) Frequency spectrum analysis of filter outputs on 13[th] layer. Left: Shared filter outputs. Center: Expert filters with spectral bias outputs. Right: Expert filters without spectral bias outputs.

