# OpenReview forum: "Graph Signal Processing Meets Mamba2: Adaptive Filter Bank via Delta Modulation"
_ICLR.cc/2026/Conference — ICLR 2026 Poster_

### Official Review · Reviewer_BvVa · 2025-10-28

**Soundness:** 3
**Presentation:** 3
**Contribution:** 3
**Rating:** 6
**Confidence:** 3

**Summary:**

The paper leverages the framework of Graph Signal Processing (GSP) to interpret and enhance Mamba2, and proposes a hierarchical filter band model architecture, HADES. Each SSM head is interpreted as a graph filter over a line graph, and HADES leverages shared and expert filters for capturing global and local properties, respectively. The paper further designed two losses - load balance loss and diversity loss - to optimize the learning of the expert filters. Empirical results show that HADES reports comparable to better accuracy than baseline across language modeling, reasoning, and retrieval, without long context fine-tuning. Filter selection analysis and frequency response analysis further validate the behavior of the expert filters.

**Strengths:**

1. The graph signal processing perspective on Mamba and ssm architecture is novel. Connecting the S4 and Mamba2 kernel to graph filters on a line graph and extending to time-varying filters makes the multi-heads as filter-banks interpretation explicit and operational. The shared plus expert filter design with spectral bias is a novel mechanism to impose functional diversity with small overhead.
2. The architecture design (shared and expert filter with respective modulations) and loss design (load balance loss to prevent collapse of filter usage and diversity loss for decorrelating filter outputs) are well-motivated, clearly formulated, and empirically supported by in-depth analysis.
3. The experimental design, especially the in-depth analysis into the behavior of the filters, provides interesting insight into the effectiveness and efficiency of the model, further validating the approach.

**Weaknesses:**

1. While the GSP-inspired perspective is intriguing, it mostly serves as an interpretation rather than a source of rigorous new theory. The idea of a ``line graph filter” for a time series is essentially equivalent to a standard 1-D convolution or recursive filter, which is a well-studied concept. The paper does not derive new analytical results (e.g. no formal theorem about stability or frequency response), and it doesn’t seem to leverage graph signal processing theory beyond the descriptive level.
2. The HADES model introduces additional complexity over the base Mamba2. It splits the SSM heads into shared vs expert categories and requires a routing network per time step, along with new hyperparameters.
3. The comparison only consists of baseline models (mamba1, Mamba2, and linear transformer variants). It would be interesting to see the performance comparison between HADES and Mamba2 models with additional comparable enhancements, such as mixture models.

**Questions:**

1. How is the training speed and convergence behavior of HADES compared to baseline models?
2. How are the gradients of the Top-K expert selection propagated?

---

> ### Author Response · Authors · 2025-11-22
> **Official Comment by Authors (1/2)**
>
> Thank you for your insightful and constructive feedback. In this response, we address each of your comments in detail and provide additional experimental results and clarifications.
>
> ## **1. GSP is not used beyond the descriptive level (W1)**
> > **Original Comment**: **W1.** While the GSP-inspired perspective is intriguing, it mostly serves as an interpretation rather than a source of rigorous new theory. The idea of a ``line graph filter” for a time series is essentially equivalent to a standard 1-D convolution or recursive filter, which is a well-studied concept. The paper does not derive new analytical results (e.g. no formal theorem about stability or frequency response), and it doesn’t seem to leverage graph signal processing theory beyond the descriptive level.
>
> `Revised Section: Section 1, Fig.1 / Section 4, Fig. 5 and Fig. 6 / Appendix E.3, Fig. 8`
>
> We appreciate the reviewer’s perspective and clarify that the GSP view in our work is not intended as a new formal theory, but as a principled diagnostic lens that directly motivated our architectural choices. Indeed, filtering on a line graph is mathematically equivalent to a 1-D convolution/recurrence, but the value of the analysis lies in how it exposed specific structural issues in Mamba2 and guided our design.
>
> In the revised manuscript, we present a unified set of empirical spectral analyses—including frequency response of filters (Section 4, Fig. 6), effective rank (Fig. 1), output spectrum (Section 4, Fig. 5), and CKA similarity (Appendix E.3, Fig. 8). These analyses consistently show that Mamba2’s heads largely collapse into smooth kernels, preserving mainly low-frequency information and producing highly redundant outputs. This behavior indicates that Mamba2 does not naturally learn meaningful spectral decomposition across heads.
>
> These findings directly motivated our architecture: we introduce shared filters that capture global information and expert filters whose $\Delta$-modulated dynamics respond to high-frequency residuals. Although we avoid explicit DSP/GSP operators for efficiency, this design is grounded in the classical GSP principle of separating smooth and residual components. The learned model exhibits exactly this specialization in practice, with ripple-like expert filters, more diverse output spectra, reduced redundancy, and a markedly higher effective rank.
>
> Thus, while our use of GSP is implicit rather than theoretical, it plays a substantive role: it diagnoses the limitations of Mamba2, informs the structure of HADES, and is empirically validated by the emergent spectral behavior of the learned filters.
>
> ---
>
> ## **2. Additional Complexity over the base Mamba2 (W2)**
> > **Original Comment**: **W2.** The HADES model introduces additional complexity over the base Mamba2. It splits the SSM heads into shared vs expert categories and requires a routing network per time step, along with new hyperparameters.
>
> `Revised Section: Appendix F`
>
> In the revised manuscript, we provide a full derivation of HADES’s computational complexity in Appendix F. Here, we summarize the main points:
> * HADES consists of two parts: (1) the mixer computation and (2) the lightweight routing.
> * The mixer component mirrors Mamba2, except that HADES activates only $H$ out of the original $M$ filters. Because mixer cost scales linearly with the number of active filters, this reduces the dominant Mamba2 cost from $T \cdot M \cdot \operatorname{FLOPs}\_\text{filter} $  to $T·H·\operatorname{FLOPs}\_{\text{filter}}$, where $\mathrm{FLOPs}\_{\text{filter}}$ includes the expensive $O(N \log N)$ SSM kernel.
> * The routing operations—score projection, top-Q selection, spectral bias, and delta modulation—are absent in Mamba2 but extremely inexpensive. Each is only $O(d)–O(H)$, several orders of magnitude cheaper than the SSM mixer. Their exact per-token FLOPs are listed in Appendix F.5 (Tables 13–14).
> * Putting these together, the total cost satisfies
>     $$\mathrm{FLOPs}\_{\text{HADES}} = \mathrm{FLOPs}\_{\text{mixer}} + \mathrm{FLOPs}\_{\text{routing}},$$ and the reduction compared to Mamba2 is
> $$\Delta \mathrm{FLOPs} = T(M-H)\mathrm{FLOPs}\_{\text{filter}} - \mathrm{FLOPs}\_{\text{routing}}.$$ Because the mixer term dominates, we obtain
> $$\mathrm{FLOPs}\_{\text{HADES}} \approx \frac{H}{M} \cdot \mathrm{FLOPs}\_{\text{Mamba2}},$$ consistent with our measured latency.
>
> We hope this clarifies that the added routing mechanism incurs negligible cost, while the reduced number of active filters yields a substantial overall efficiency gain. All details and derivations appear in Appendix F.

---

> ### Author Response · Authors · 2025-11-22
> **Official Comment by Authors (2/2)**
>
> ## **3. Additional Baseline (W3)**
> > **Original Comment**: **W3.** The comparison only consists of baseline models (mamba1, Mamba2, and linear transformer variants). It would be interesting to see the performance comparison between HADES and Mamba2 models with additional comparable enhancements, such as mixture models.
>
> `Revised Section: Section D.2, Table 5`
>
> To address your question, we additionally compare our method against a mixture variant, MoM [1], the mixture-of-experts extension of Gated DeltaNet [2], itself an adaptation of Mamba2. Across all benchmarks, our model consistently achieves comparable or higher average performance compared to MoM, demonstrating the effectiveness of our approach even relative to mixture-based variants.
>
> **Table 5. Performance comparison on language modeling and zero-shot common-sense reasoning**
> | Tasks | Wikitext | Lambada | Lambada |  PiQA  | hellaswag |  Wino. |  ARC_E |  ARC_C |  BoolQ | OpenbookQA |  Avg.  |
> |-|-|-|-|-|-|-|-|-|-|-|-|
> | Metric |ppl |   ppl   |   acc   |   acc  |   acc_n   |   acc  |   acc  |  acc_n |   acc  |    acc_n   |        |
> | MoM | 31.58 | 23.28 | **40.40** | 62.57 | **32.99** | **52.64** | 44.91 | **23.89** | 52.78 | 27.20 | 42.17 |
> | *HADES* | **31.48** | **21.74** | 39.24 | **63.93** | 32.82 | **52.64** | **45.03** | 22.01 | **58.84** | **28.80** | **42.91** |
>
> ---
>
> ## **4. Training speed and convergence behavior of HADES (Q1)**
> > **Original Comment**: **Q1.** How is the training speed and convergence behavior of HADES compared to baseline models?
>
> `Revised Section: Appendix F.6, Figure 11`
>
> To address your question, we added loss convergence plot and training speed plot of HADES and Mamba2 in the revised manuscript (Appendix F.6, Figure 11).
>
> We also provide the table below for your clarity:
>
> **Table 15. Training time analysis of average compute time measured over a 100-step window in one epoch training regime.**
> | Model  |  Mean Step Time (s) |
> |-|-|
> | Mamba1      |   143.0105            |
> | Mamba2      | 104.0216            |
> | *HADES*      |  96.5639             |
>
> From these results, we observe that HADES converges stably and at a similar rate to the baselines, and achieves step time comparable to Mamba2.
>
> ---
>
> ## **5. Gradient propagation of the Top-K expert selection (Q2)**
> > **Original Comment**: **Q2.** How are the gradients of the Top-K expert selection propagated?
>
> `Revised Section: None`
>
> Our routing mechanism uses hard Top-k selection, and the indices produced by the top-k operator are non-differentiable. As a result, gradients do not flow through the selection decisions themselves. However, once the experts are selected via `gather`, the corresponding selected $\Delta$ values do receive gradients normally, while unselected experts simply do not receive gradients for that step. Although the routing choice is non-differentiable, the router features $h$ are still updated through the auxiliary MoE-style load-balancing loss. This auxiliary objective provides a learning signal, allowing it to refine its selection behavior even without gradients propagating through the top-k operator.
>
> In short, this training dynamic is consistent with standard hard-routing MoE approaches: the routing decision is discrete, but both the experts and the router features continue to receive meaningful gradients through their respective learning signals.
>
> For clarity, the forward/backward gradient flow is summarized below:
>
> ```
> Forward:
> h → topk → select_ids → gather → dt (selected) → SSM → output
>
> Backward:
> output → SSM → dt (selected) → gather → dt (original)
>                                      ↓
>                                      h (through auxiliary losses)
> ```
>
> ---
>
> ## References
> [1] Du et al., MoM: Linear Sequence Modeling with Mixture-of-Memories, arXiv 2025
>
> [2] Yang et al., Gated Delta Networks: Improving Mamba2 with Delta Rule, ICLR 2025

---

> > ### Author Response · Authors · 2025-11-26
> >
> > Dear Reviewr BvVa,
> >
> > Thank you for your detailed and considerate review of our paper. Your comments were extremely helpful, and we carefully incorporated your suggestions and clarifications into our responses throughout the discussion period. We would be grateful if you could take a moment to revisit our replies and reassess your rating in light of the additional explanations provided.
> >
> > We truly appreciate your time and effort. Please do not hesitate to contact us if anything else needs clarification.
> >
> > Warm regards,
> > Authors

---

### Official Review · Reviewer_neR9 · 2025-11-01

**Soundness:** 3
**Presentation:** 4
**Contribution:** 2
**Rating:** 4
**Confidence:** 4

**Summary:**

The paper introduces a novel SSM-based architecture that builds upon Mamba2. Hierarchical ADaptive filter bank for Efficient SSMs (HADES) is a Graph Signal Processing (GSP)-inspired framework that modifies portions of Mamba2, such that each head corresponds to a graph filter (here, the sequence of tokens is viewed as a line graph) and the multi-head SSM to a bank of filter whose outputs are aggregated. HADES introduces two kinds of filter: namely, a shared filters that is applied (always selected so not part of routing) to capture global content, and expert filters that are routed per token (similar to Mixture-of-Experts or Mixture-of-Heads). Here, the router routes based on the "spectral residual", defined as $x_t - mean(x_1,...,x_t)$, and $\Delta_{t, base}$. In order to guarantee that all expert filters are used evenly, they add a load-balancing loss, similar to MoE designs, and a diversity loss that penalizes filter outputs' deviations from pairwise orthogonality.

Experimentally, the authors show that HADES outperforms other architectures in this line of work, such as Mamba2 and DeltaNet, on 8 zero-shot language tasks (those standard in LM-eval harness and used in the Mamba2 paper for evaluation). When trained on 200B tokens on Pile, their 218M-parameter model outperforms 370M-parameter baseline models. Plus, there is no issue regarding latency and computation memory. Hades beats Mamba2 in long-context retrieval (pass-key retrieval). Finally, the paper offers some interpretability analyses. Frequency‑domain plots suggest shared filters act low‑pass while expert filters emphasize higher frequencies, particularly with the spectral bias on $\Delta$, and expert‑selection heatmaps show task‑region‑specific specializations.

**Strengths:**

Originality:
- Recasts multi-head Mamba2 as a graph filter bank on a line graph, connecting LTV SSMs to graph signal processing (GSP) and framing heads as node-variant graph filters. Introduces a novel architecture HADES, a hierarchical filter bank with (i) always-on shared filters and (ii) token-routed expert filters, selected via a spectral residual and $\Delta$-modulation.
- Their construction of expert filters creates more opportunity for modular/interpretable filters, as they are trained to be distinct from each other.

Quality:
- Strong efficiency–performance trade-off: ~59% of Mamba2’s parameters (≈218M vs. 370M) while matching or surpassing baselines on zero-shot reasoning and long-context passkey retrieval; reported ~1.37× speed and memory gains.
- Their construction is not complicated and does not add much computational overhead when routing.
- The ablation and sensitivity analyses make the empirical investigation more thorough, and show that each of their designs are necessary.

Clarity:
- The presentation is clear for readers who are less familiar with SSM-based architecture, and the differences between Mamba2 and HADES are clear, both mathematically and visually. Background sections concisely link SSM kernels, convolutional views, and GSP operators, helping readers follow the reinterpretation.
- The hyperparameters and experimental descriptions are quite thorough, but certain sections of the appendix are not properly linked to the main body so, at times, it may be a slightly confusing.

Significance:
- While I don't find the connection entirely illuminating, the GSP lens offers a principled design template for structured, adaptive filtering in SSMs, which may be useful beyond HADES for designing and analyzing SSM-based models.
- Similarly to the points made for originality, the paper demonstrates decent empirical performances and long-context understanding ability. The devised architecture may help future interpretability research as well.

**Weaknesses:**

While I enjoyed the paper's presentation and ideas overall, I think the major weakness of the paper is the strength of their empirical evidence. I will list this in two major axes:
1. **Lack of scale**: Despite the thorough experiments in ablation, sensitivity, and multiple baselines, the paper only operates on the 200B-token Pile training of 370M-parameter models. Because we only get one data point in the (number of tokens trained, model scale), it is hard to know if the model will scale or holds in any other regime of training. If we have another model size (preferably on the larger side) and another training size (100B or 3-400B tokens), the authors would be able to make more robust conclusions.
- Also, while I acknowledge that SSM-based works such as Mamba pretrain on Pile, this practice holds less value in 2025. We have DCLM, FineWeb, and Nemotron-CC, which are better natural language datasets, but to continue doing experiments on Pile dataset seems outdated.
- Instead of the 218M HADES model, I would like to see a 370M (or FLOPs-equivalent) HADES model being compared to the baseline models to observe the real gap when controlling for FLOPs or parameters.
- "even with a drastically reduced model size of approximately 38.64% (143M) in the H = 8 setting, our model maintains performance comparable to the optimal hyperparameter configuration and even outperforms it on two tasks" (L903-905): while the authors analyze their results with such framing, it makes me wonder if HADES does not scale well due to this result. If more FLOPs or more parameters lead to better performance (a necessary feature for language models), then this result seems unintuitive.
2. **Randomness across seeds**: Table 1 and Table 3 do not account for randomness across training runs/seeds. While standard error is low, this only accounts for sampling variance from the dataset and not the variance in training. If possible, averaging evaluation accuracies across five identically trained models (different seeds) for Table 1/3 would make the results more trustworthy.
- Did you also perform a hyperparameter sweep/tuning for the baseline models, or only perform hparam tuning for HADES?

My score is between 4 and 6, so for now, I will keep my score as 4 (marginally below the acceptance threshold). Once some questions and concerns are addressed, I am willing to increase.

**Questions:**

For questions below are minor. Addressing the weaknesses (and suggestions therein) would help me re-evaluate this work.

**Minor misc.**
1. Eq. (1) in L88 should have $\bar{A} h(t)$ instead of $\bar{A} h(t-1)$, right? In continuous time one expects no shift and  $\bar{A} h(t)$ is the one written for the Mamba paper.
2. Top‑K selection is non‑differentiable; the paper does not state whether it uses straight‑through, Gumbel‑TopK, or train with continuous weights and harden at eval. Or does Top-Q selection mean something else than I am missing?
3. I don't entirely understand the load-balance loss in Eq (15). Is this written correctly and precisely? What expectation are we taking and how do we square a vector?
3. Appendix B.1 says: “For fair comparison, all models are trained... with 370M parameters...”, but I assume here HADES is 218M? It's written confusingly here. If you put the number of parameters and also FLOPs on the table, that would be helpful.
4. L1025 calculation is wrong: 368,346,624−150,407,424=217,939,200.
5. Can you also report training perplexity in Table 1?
6. Can you report the model size in Table 6? I would interested in the latency of HADES when controlling for model size or memory, but here, the latency primarily seems to come from being smaller.

---

> ### Author Response · Authors · 2025-11-22
> **Official Comment by Authors (1/3)**
>
> ## **1. Lack of scale (W1)**
>
> ### **1-1. Experiments on different training regime**
>
> > **Original Comment**:
> **W1.** Despite the thorough experiments in ablation, sensitivity, and multiple baselines, the paper only operates on the 200B-token Pile training of 370M-parameter models. Because we only get one data point in the (number of tokens trained, model scale), it is hard to know if the model will scale or holds in any other regime of training. If we have another model size (preferably on the larger side) and another training size (100B or 3-400B tokens), the authors would be able to make more robust conclusions. Also, while I acknowledge that SSM-based works such as Mamba pretrain on Pile, this practice holds less value in 2025. We have DCLM, FineWeb, and Nemotron-CC, which are better natural language datasets, but to continue doing experiments on Pile dataset seems outdated.
>
> `Revised Section: Appendix D.2, Table 4`
>
> We thank you for raising concerns about whether HADES scales beyond the 370M Pile-pretrained setting. We agree that validating our method at larger scale is crucial for assessing its generality and practical impact. To directly address this, we include results from a 1B-paramter setting, trained on the FineWeb-Edu 30B. We have included these results in the revised manuscript (Appendix D.2, Table 4). For your convenience, we also provide the same results below:
>
> **Table 4. Performance comparison on language modeling and zero-shot common-sense reasoning**
> | Tasks | Wikitext | Lambada | Lambada |  PiQA  | hellaswag |  Wino. |  ARC_E |  ARC_C |  BoolQ | OpenbookQA |  Avg.  |
> |-|-|-|-|-|-|-|-|-|-|-|-|
> | Metric | ppl |   ppl   |   acc   |   acc  |   acc_n   |   acc  |   acc  |  acc_n |   acc  |    acc_n   |        |
> | RetNet (1.3B)|  22.45 | 21.84 |  38.70 | 69.04 |    47.73 | 52.72 | 63.68 | 33.36 | 60.61 |     36.60 | 50.31 |
> | Mamba2 (1.3B)|  **19.47** | 17.40 | 40.68   | 70.29  | **53.24** | 56.04  | **69.87**  | **36.35**  | 55.81  | 37.40   | 52.46  |
> | DeltaNet (1.3B)|  19.77 | **16.64** | **41.78**   | 70.95  | 51.09  | 54.70  | 67.63  | 34.47  | **61.19**  | 38.40   | 52.53  |
> | *HADES* (1B) |  20.41 | 17.22 |  41.18   | **71.33**  | 51.85  | **56.35**  | 68.48  | 34.81  | 60.73  | **38.60**   | **52.92** |
>
> ### **1-2. 370M-matched Experiment**
> > **Original Comment**:
> **W1.** Instead of the 218M HADES model, I would like to see a 370M (or FLOPs-equivalent) HADES model being compared to the baseline models to observe the real gap when controlling for FLOPs or parameters.
>
> `Revised Section: None`
>
> Regarding the 370M-matched comparison, we emphasize that this configuration is not a standard architectural comparison. To match the parameter count of the 370M baseline, all filters are activated, instead of activating only selected filters. This setting remains below the performance of HADES (218M). We attribute this to task-dependent sparsity, and increasing the number of experts does not necessarily improve performance—an effect analogous to the well-known behavior observed in MoE models.([1])
>
> Importantly, HADES exhibits normal scaling behavior at larger model sizes. In the 1B setting (Table 4), where all models share comparable depth and width, HADES consistently outperforms its corresponding baseline. This demonstrates that HADES continues to benefit from increased capacity in the same way standard language models do, while being more parameter-efficient at small scales.
>
> **Table 15. 370M-matched Experiment Result**
> | Model | Wiki.   | LMB.|LMB.| PIQA  | Hella. | Wino.   | ARC-e  | ARC-c   | BoolQ    | OBQA.  | Avg.    |
> |-|-|-|-|-|-|-|-|-|-|-|-|
> | Metric | ppl |   ppl   |   acc   |   acc  |   acc_n   |   acc  |   acc  |  acc_n |   acc  |    acc_n   |        |
> | HADES (370M)  | 33.33 | 26.44 | 37.30 | 63.17    | 31.86      | 50.20 | 43.27 | 23.21 | 61.07 | 29.20 | 42.41 |
>
> ### **1-3. Concerns on Scaling Behavior**
> > **Original Comment**:
> **W1.** "even with a drastically reduced model size of approximately 38.64% (143M) in the H = 8 setting, our model maintains performance comparable to the optimal hyperparameter configuration and even outperforms it on two tasks" (L903-905): while the authors analyze their results with such framing, it makes me wonder if HADES does not scale well due to this result. If more FLOPs or more parameters lead to better performance (a necessary feature for language models), then this result seems unintuitive.
>
> `Revised Section: None`
>
> We agree that the 8-filter configuration occasionally outperforming the 16-filter variant may seem counterintuitive, but we do not believe this indicates a scaling limitation. Such behavior naturally arises from task-dependent sparsity effects: with only eight filters, the router often forms more specialized and stable expert assignments that align well with certain datasets. This variability is expected in small, highly sparse models and should not be construed as a structural scaling issue.

---

> ### Author Response · Authors · 2025-11-22
> **Official Comment by Authors (2/3)**
>
> ## **2. Regarding Robustness across seeds & Hyperparameter Search (W2)**
> ### **2-1.Regarding Robustness across seeds**
> > **Original Comment**: **W2.** Table 1 and Table 3 do not account for randomness across training runs/seeds. While standard error is low, this only accounts for sampling variance from the dataset and not the variance in training. If possible, averaging evaluation accuracies across five identically trained models (different seeds) for Table 1/3 would make the results more trustworthy.
>
> `Revised Section: Appendix D.2, Table 6`
>
> As requested, we aimed to perform a full seed sweep for all baselines; however, due to resource limitations, conducting exhaustive searches across every model was not feasible. Instead, we focused on the most relevant comparison: our primary baseline, Mamba2, which also achieved the second-best average performance.
>
> For both our model and Mamba2, we performed searches over three random seeds and report the mean and standard deviation in our revised manuscript (Appendix D.2, Table 6). For convenience, we also reproduce the corresponding table below. *HADES* consistently outperform Mamba2 on average score over all eight benchmarks.
>
> **Table 6. Performance comparison on language modeling and zero-shot common-sense reasoning**
> | Model | Wiki.   | LMB.|LMB.| BoolQ  | Hella. | Wino.   | ARC-e  | ARC-c   | PIQA    | OBQA.  | Avg.    |
> |-|-|-|-|-|-|-|-|-|-|-|-|
> |Mamba2 (Avg.) | 30.64|22.57|	37.85|54.63|33.27|51.15|45.34|23.29|63.04|27.93|42.06|
> |(Std)|0.752|1.662|0.014|0.007|0.007|0.006|0.004|0.009|0.004|0.014|0.008|
> |HADES (Avg.) |33.41|25.57|37.10|59.76|31.70|51.85|44.84|22.67|63.15|27.87|42.37|
> |(Std)| 1.721|3.482|	0.022|0.009|0.010|0.011|0.008|0.006|0.007|0.013|0.005|
>
> We additionally provide mean and standard deviation results for Table 3 following the same procedure in Table 17 here. The first line shows the mean over three seeds with (seed std); the second line shows the evaluation standard error with (seed std).
>
> **Table 17. Performance comparison on language modeling and zero-shot common-sense reasoning with evaluation standard error**
> | Model | Wiki.|LMB.|LMB.| BoolQ|Hella.| Wino.|ARC-e| ARC-c| PIQA| OBQA.|Avg.|
> |-|-|-|-|-|-|-|-|-|-|-|-|
> |Mamba2 | 30.64 (0.752)|22.57 (1.662)|	37.85 (0.014)|54.63 (0.007)|33.27 (0.007)|	51.15 (0.006)|45.34 (0.004)|23.29 (0.009)|	63.04 (0.004) |	27.93 (0.014)|	42.06 (0.008)|
> | Eval. Std| N/A | 0.747 (0.066) | 0.007 (0.000) | 0.009 (0.000) | 0.005 (0.000) | 0.014 (0.000) | 0.010 (0.000) | 0.012 (0.000) | 0.011 (0.000) | 0.020 (0.000) | 0.011 (0.000) |
> | HADES (Ours) | 33.41 (1.721) | 25.57 (3.482) | 37.10 (0.022) | 59.67 (0.009) | 31.70 (0.010) | 51.85 (0.011) | 44.84 (0.008) | 22.67 (0.006) | 63.15 (0.007) | 27.87 (0.013) | 42.37 (0.011) |
> | Eval. Std| N/A | 0.877 (0.135) | 0.007 (0.000) | 0.009 (0.000) | 0.005 (0.000) | 0.014 (0.000) | 0.010 (0.000) | 0.012 (0.000) | 0.011 (0.000) | 0.020 (0.000) | 0.011 (0.000) |
>
> ### **2-2. Hyperparameter Search**
> > **Original Comment**: **W2.** Did you also perform a hyperparameter sweep/tuning for the baseline models, or only perform hparam tuning for HADES?
>
> `Revised Section: None`
>
> For the other baseline models, we relied on their publicly reported settings and performed a learning-rate tuning, without conducting additional extensive hyperparameter searches due to resource constraints.
>
> ---
>
> ## **3. Top-K selection is non-differentiable (Q2)**
> > **Original Comment**: Top‑K selection is non‑differentiable; the paper does not state whether it uses straight‑through, Gumbel‑TopK, or train with continuous weights and harden at eval. Or does Top-Q selection mean something else than I am missing?
>
> `Revised Section: None`
>
> Our routing mechanism uses hard top-k selection, and the indices produced by the top-k operator are non-differentiable. As a result, gradients do not flow through the routing decisions themselves. However, once the experts are selected via `gather`, the corresponding expert outputs (e.g., delta values) do receive gradients normally, while unselected experts simply do not receive gradients for that step. Although the routing choice is non-differentiable, the router features $h$ are still updated through the auxiliary MoE-style load-balancing loss. This auxiliary objective provides a learning signal to the router, allowing it to refine its selection behavior even without gradients propagating through the top-k operator.
>
> In short, this training dynamic is consistent with standard hard-routing MoE approaches: the routing decision is discrete, but both the experts and the router features continue to receive meaningful gradients through their respective learning signals.
>
> For clarity, the forward/backward gradient flow is summarized below:
> ```
> Forward:
> h → topk → select_ids → gather → dt (selected) → SSM → output
>
> Backward:
> output → SSM → dt (selected) → gather → dt (original)
>                                      ↓
>                                      h (through auxiliary losses)
> ```

---

> ### Author Response · Authors · 2025-11-26
> **Official Comment by Authors (3/3)**
>
> ## **4. Load-Balance Loss in Eq.15 (Q3)**
>
> > **Original Comment**: **Q3.** I don't entirely understand the load-balance loss in Eq (15). Is this written correctly and precisely? What expectation are we taking and how do we square a vector?
>
> `Revised Section: None`
>
> Thank you for the question regarding Eq. (15). The load-balance loss is computed exactly as the squared coefficient of variation over the expert selection scores. Here, $s_t$ is a vector containing the selection scores for all experts at time step t. The expectation $\mathbb{E}[s_t]$ refers to the mean of this vector taken across the expert dimension, and the variance $\mathrm{Var}(s_t)$ is computed over the same dimension. Our implementation follows this formulation directly by applying the CV squared computation to the selection score $s_t$, which is produced by passing the spectral-residual and $\Delta_t$ features through the selection projection. This ensures the loss is well-defined, mathematically precise, and aligned with the intended role of encouraging balanced expert utilization. We provide pseudo code below:
>
> ~~~python
> Function LOAD_BALANCE_LOSS(x):
> # Computes the squared coefficient of variation of a sample.
> # Returns 0 if the input has only one element.
>
> eps ← 1e-10
>
> If length(x) = 1:
>     Return 0
>
> variance ← VARIANCE(x)
> mean_val ← MEAN(x)
>
> cv_sq ← variance / (mean_val^2 + eps)
>
> Return cv_sq
> ~~~
>
> ---
>
> ## **5. Training Perplexity (Q6)**
>
> > **Original Comment**: **Q6.** Can you also report training perplexity in Table 1?
>
> `Revised Section: Table 1`
>
> To address your question, we provide training perplexity in our revised manuscript (Table 1). HADES showed reasonable training perplexity compared to baselines.
>
> We also provide the table below for your clarity:
>
> **Table 1. Final train perplexity across models.**
> | Model  | Train Perplexity |
> |-|-|
> | Linear Attention |  2.49   |
> | RetNet      |  2.41   |
> | Mamba1      |  2.53   |
> | Mamba2      |  2.33   |
> | DeltaNet    |  2.29   |
> | HADES      |  2.31   |
>
> ---
>
> ## **6. Minor revisions**
>
> > **Original Comment**:
> **Q1.** Eq. (1) in L88 should have $\bar{A}h(t)$ instead of $\bar{A}h(t-1)$, right? In continuous time one expects no shift and $\bar{A}h(t)$ is the one written for the Mamba paper.
> **Q4.** Appendix B.1 says: “For fair comparison, all models are trained... with 370M parameters...”, but I assume here HADES is 218M? It's written confusingly here. If you put the number of parameters and also FLOPs on the table, that would be helpful.
> **Q5.** L1025 calculation is wrong: 368,346,624−150,407,424=217,939,200.
> **Q7.** Can you report the model size in Table 6? I would interested in the latency of HADES when controlling for model size or memory, but here, the latency primarily seems to come from being smaller.
>
> `Revised Section: Section 2, Eq.1 / Appendix F.1, Table 9 / Appendix F.4`
>
> We appreciate the reviewer’s detailed examination of our manuscript. All minor errors noted in the review have been corrected, including the typo in Eq. (1) (Q1), clarification in Appendix B.1 (now C.1) (Q4), the miscalculation reported in (Q5), and addition of. parameter count in Table 6 (now 8) (Q7)
>
> ---
>
> ## References
> [1] Dai et al., DeepSeekMoE: Towards Ultimate Expert Specialization in Mixture-of-Experts Language Models, ACL 2024

---

> > ### Author Response · Authors · 2025-11-26
> >
> > Dear Reviewer neR9,
> >
> > We wish to sincerely thank you for the thoughtful and meticulous review you provided. Your feedback has been extremely helpful in guiding us to further improve our work. Throughout the discussion period, we have endeavored to address each of your comments with the utmost care and clarity. If it would be possible, we would be very grateful if you could review our responses and kindly reconsider your assessment in light of the additional details we have provided.
> >
> > We deeply appreciate your time, effort, and valuable insights. Please feel free to let us know if any further clarification is needed.
> >
> > With sincere gratitude,
> > Authors

---

> > > ### Comment · Reviewer_neR9 · 2025-11-27
> > > **Re-evaluation**
> > >
> > > Thank you for the descriptive response. I appreciate all the details and experiments you have additionally ran.
> > > I have re-evaluated and have increased my score to a 6.

---

### Official Review · Reviewer_QPLB · 2025-11-01

**Soundness:** 3
**Presentation:** 3
**Contribution:** 3
**Rating:** 6
**Confidence:** 3

**Summary:**

The paper analyzes multi-head SSMs and Mamba2 as a graph filter bank over a line graph, and introduces HADES. HADES routes per-token to a small set of expert filters, while shared filters are always applied to preserve global information. HADES adds two regularizers - (i) load-balance over expert scores and (ii) output-diversity across filters to prevent collapse. HADES reports competitive performance across 8 zero-shot tasks while using only 58.9% of the parameters, and demonstrate stronger passkey retrieval upto 16K context length.

**Strengths:**

- The formalization of SSM heads into Graph Signal Processing's filter bank is clear and allows principled analysis about low-pass and adaptive behavior.
- The routing/bias mechanism tied to \Delta_{HADES} gives a minimal hook for content-adaptive dynamics which aligns with SSM parametrization.
- Competitive results at a much lower parameter count regime and performance improvements in long-context tasks.
- The ablation study and analysis are thorough.

**Weaknesses:**

- While the competence of HADES with respect to the reduced number of parameters seems promising, the architecture does seem to cause more FLOP overhead. Listing this analysis would strengthen the contributions of this work.
- The spectral analysis (FFT) on hidden sequences from one layer may be confounded with the layer or the gamma value.

**Questions:**

- For passkey, can the region order be randomized to evaluate the model's robustness?
- When computing the running mean for the token router, is this strictly causal?
- How might HADES compare to simpler head routing apporaches under the same budget?

---

> ### Author Response · Authors · 2025-11-22
> **Official Comment by Authors (1/2)**
>
> Thank you for your insightful and constructive feedback. In this response, we address each of your comments in detail and provide additional experimental results and clarifications.
>
> ## **1. Regarding FLOPs Overhead (W1)**
>
> > **Original Comment**: **W1.** While the competence of HADES with respect to the reduced number of parameters seems promising, the architecture does seem to cause more FLOP overhead. Listing this analysis would strengthen the contributions of this work.
>
> `Revised Section: Appendix F, Table 13-14`
>
> In the revised manuscript, we provide a full derivation of HADES’s computational complexity in Appendix F. Here, we summarize the main points:
> * HADES consists of two parts: (1) the mixer computation and (2) the lightweight routing.
> * The mixer component mirrors Mamba2, except that HADES activates only $H$ out of the original $M$ filters. Because mixer cost scales linearly with the number of active filters, this reduces the dominant Mamba2 cost from $T·M·\mathrm{FLOPs}\_{\text{filter}}$ to $T·H·\mathrm{FLOPs}\_{\text{filter}}$, where $\mathrm{FLOPs}\_{\text{filter}}$ includes the expensive $O(N \log N)$ SSM kernel.
> * The routing operations—score projection, top-Q selection, spectral bias, and delta modulation—are absent in Mamba2 but extremely inexpensive. Each is only $O(d)–O(H)$, several orders of magnitude cheaper than the SSM mixer. Their exact per-token FLOPs are listed in Appendix F.5 (Tables 13–14).
> * Putting these together, the total cost satisfies
>     $$\mathrm{FLOPs}\_{\text{HADES}} = \mathrm{FLOPs}\_{\text{mixer}} + \mathrm{FLOPs}\_{\text{routing}},$$ and the reduction compared to Mamba2 is
> $$\Delta \mathrm{FLOPs} = T(M-H)\mathrm{FLOPs}\_{\text{filter}} - \mathrm{FLOPs}\_{\text{routing}}.$$ Because the mixer term dominates, we obtain
> $$\mathrm{FLOPs}\_{\text{HADES}} \approx \frac{H}{M} \cdot \mathrm{FLOPs}\_{\text{Mamba2}},$$ consistent with our measured latency.
>
> We hope this clarifies that the added routing mechanism incurs negligible cost, while the reduced number of active filters yields a substantial overall efficiency gain. All details and derivations appear in Appendix F.
>
> ---
>
> ## **2. Regarding confound spectrum analysis (W2)**
>
> > **Original Comment**:
> **W2.** The spectral analysis (FFT) on hidden sequences from one layer may be confounded with the layer or the gamma value.
>
> `Revised Section: Appendix H, Figure 12`
>
> We appreciate the reviewer’s concern about potential confounders in our spectral analysis. We would like to clarify how the experiments in Fig. 5 were conducted. First, all spectra in Fig. 5 are computed from the same layer (13th) and under the same set of hyper-parameters, including the global $\gamma$ in Eq. (14). The only differences between panels are (i) whether the filter is shared vs. expert, and (ii) whether the spectral-bias term $f_b([\Delta_{t,\text{base}} \| r_t])$ is enabled for the expert filters. In addition, we apply FFT to the normalized filter outputs (“Normalized Magnitude” in the y-axis), so global rescaling effects from $\gamma$ do not change the relative frequency profile.
>
> We repeated the FFT analysis on earlier and later layers, which is already in Appendix E.4 Fig. 9 (12th, 29th and 45th layers). In all cases we observe the similar qualitative pattern: shared filters behave as low-pass filters, while expert filters with spectral bias exhibit certain effects on frequency components than expert filters without spectral bias.
>
> Consistently, in a $\gamma$-sweep performed at inference time on the same trained model, varying $\gamma$ alone does not meaningfully change the expert filters’ frequency spectrum (Appendix H, Fig. 12(a)), whereas the presence or absence of the bias yields markedly different spectral patterns (Fig. 12(b) and 12(c)). These results confirm that the change of frequency spectrum observed in Fig.7(b) and Fig.7(c) is attributable to the residual-based bias network $f_b$, rather than to layer or $\gamma$ effects.

---

> ### Author Response · Authors · 2025-11-22
> **Official Comment by Authors (2/2)**
>
> ## **3. Passkey Robustness (Q1)**
> > **Original Comment**:
> **Q1.** For passkey, can the region order be randomized to evaluate the model's robustness?
>
> `Revised Section: None`
>
> As described in Section 4.2 and Appendix C.2, the passkey prompt consists of four semantic regions: Task Description, Passkey, Query, and Dummy Text.
>
> First, we note that the ordering of these semantic regions follows the standard evaluation protocol used in prior work [1], and our setup remains consistent with this established benchmark design.
>
> If your question instead concerns robustness with respect to the position of the passkey within the dummy region, our evaluation directly addresses this aspect. For the passkey experiments, the prompt is constructed by inserting the key at specific depth locations within the dummy text (e.g., 10%, 20%, …), allowing us to systematically analyze performance changes across different key positions.
>
> For each depth, we conduct 10 independent trials, and the Depth Percent axis in Fig. 3 corresponds to these predefined depth locations. Each data point in Fig. 3 therefore represents the average performance over 10 independent evaluations at that depth. Because all trials are conducted independently, this experimental setup provides a reliable and robust assessment of the model’s passkey-retrieval capability across varying key positions.
>
> ---
>
> ## **4. Causality of Running Mean (Q2)**
> > **Original Comment**:
> **Q2**. When computing the running mean for the token router, is this strictly causal?
>
> `Revised Section: Section 3.1`
>
> Yes. The running mean used in the token router is strictly causal. Because the prefill and decode phases operate differently (processing many tokens at once vs. one token at a time), we use two causal implementations while preserving the same semantics. We include description and pseudocode for each phase below for clarity. Also, We have added a short clarification of the running mean $\mu$ in the router in the revised manuscript to avoid any ambiguity.
>
> ### **4-1. Training / Prefill (causal cumulative mean)**
>
> During prefill phase, we compute the running mean using a causal cumulative-sum formulation: the cumulative sum at step $t$ includes only tokens $1:t$. This guarantees that no future information is ever used during prefill.
>
> ~~~python
> # Inputs:
> #   u: input sequence of length L
> # Output:
> #   spectral_residual
>
> cumsum[0] = 0
> for t = 1 to L:
>     cumsum[t] = cumsum[t-1] + u[t]         # cumulative sum
>     mean[t] = cumsum[t] / t                # running mean at position t
>
> spectral_residual = u - mean
> ~~~
>
> ### **4-2. Inference / Decode (cached causal update)**
> During autoregressive decoding, we cache the cumulative sum from previous steps and update it in-place whenever a new token arrives. The running mean at timestep $t$ is therefore computed solely from the cached past sum and the current token.
>
>
> ~~~python
> # Cache states:
> #   conv_state, ssm_state, cumsum_state
> # Inputs:
> #   u: current input token
> #   t_pos: current time step (1-indexed)
>
> # Load cached states
> conv_state, ssm_state, cumsum_state = load_from_cache()
>
> # Update cumulative sum with new token
> cumsum_state = cumsum_state + u
>
> # Running mean for current step
> running_mean = cumsum_state / (t_pos)
>
> # Compute residual
> spectral_residual = u - running_mean
> ~~~
>
> Since both phases rely only on $u\_{1:t}$, the router’s running-mean computation remains strictly causal under all evaluation modes.
>
> ---
>
> ## **5. Comparison to Simpler Head Routing Under Same Budget (Q3)**
> > **Original Comment**:
> **Q3.** How might HADES compare to simpler head routing apporaches under the same budget?
>
> `Revised Section: Appendix E.2, Table 7`
>
> To address your concern, we did additional ablation study on router mechanisms. We add 2 types of routers : Gumbel Softmax Top-k router and Input linear projection router. Full resuls are included in the revised manuscript (Appendix E.2, Table 6). Our GSP router showed superior performance over both routers; indicating GSP analogy performs well. We also provide summarized table below:
>
> **Table 6.1 Ablation on filter selection routing**
> | Tasks |Parameter Reduction| Wikitext | Lambada | Lambada |  PiQA  | hellaswag |  Wino. |  ARC_E |  ARC_C |  BoolQ | OpenbookQA |  Avg.  |
> |-|-|-|-|-|-|-|-|-|-|-|-|-|
> | Metric |-| ppl |   ppl   |   acc   |   acc  |   acc_n   |   acc  |   acc  |  acc_n |   acc  |    acc_n   |        |
> | HADES |-| 31.48 | 21.74 |39.24 |63.93| 32.82 |52.64 |45.03 |22.01 |58.84 |28.80 |42.91|
> | Random Routing |-0K| 35.78 | 32.77 | 33.17 | 61.97 | 30.47 | 52.49 | 43.31 | 23.12 | 55.72 | 28.00 | 41.03 |
> | Input-only (Linear routing)|-3K|	34.17|	23.95	| 38.40  | 63.71  | 31.60  | 51.22  | 43.56  | 23.04  | 58.35  | 27.80  | 42.21  |
> | Gumbel Softmax Top-K Routing	|-0K|34.83| 27.21 | 36.95  | 62.19  | 31.39  | 50.51  | 43.43  | 22.35  | 58.07  | 28.20  | 41.64  |
>
> ---
>
> ## References
> [1] Chen et al. Longlora: Efficient fine-tuning of long-context large language models. ICLR, 2024.

---

> > ### Author Response · Authors · 2025-11-26
> >
> > Dear Reviewer QPLB,
> >
> > We are sincerely grateful for the time and care you invested in reviewing our work. Your comments were highly valuable to us, and we have done our utmost to address each of them thoroughly during the discussion period. If it is not too much to ask, we would be deeply appreciative if you could take a moment to read our responses and kindly reconsider your evaluation in light of the clarifications we have provided.
> >
> > Thank you again for your thoughtful feedback and your contribution to improving our paper. Please let us know if there is anything further we can clarify.
> >
> > With sincere appreciation,
> > Authors

---

### Official Review · Reviewer_kLjo · 2025-11-04

**Soundness:** 3
**Presentation:** 2
**Contribution:** 2
**Rating:** 4
**Confidence:** 5

**Summary:**

This paper proposes HADES, a new architecture based on Mamba2 that is inspired by Graph Signal Processing (GSP). The authors frame Mamba2's multi-head structure as a filter bank operating on a line graph. The HADES model introduces a hierarchy of filters: shared filters for global, low-frequency information and expert filters for local, high-frequency details. A routing mechanism selects which expert filters to apply to a token based on its "spectral residual," which is also used to modulate the delta parameter. The method is regularized using two auxiliary losses to encourage filter balance and diversity. The primary contribution is a model that achieves comparable performance to Mamba2 on several benchmarks while using only 58.9% of the parameters.

**Strengths:**

1. The paper introduces an interesting conceptual link between Graph Signal Processing and Mamba2, re-framing multi head SSMs as filter banks. This GSP perspective motivates a novel routing mechanism based on a spectral residual which is a creative approach to token-adaptive computation.

2. The authors conduct a thorough set of ablations on their proposed 370M parameter model, which helps validate the components of their design, such as the auxiliary losses and the hierarchical filter structure. The parameter reduction to 58.9% at this scale is notable. The in-depth analysis showing filter specialization (low-pass vs. high-pass) is also a quality contribution.

3.The paper is clearly written, and the figures are helpful for understanding the architecture and its intended behavior. The authors provide a good explanation of their GSP-inspired framework.

4. The work demonstrates a potential path to more parameter-efficient SSMs. The in-depth analysis showing filter specialization is a good step towards more interpretable models.

**Weaknesses:**

1. The paper's primary weakness is that all experiments are confined to a single, small 370M parameter model. The central claim of 58.9% parameter savings is not validated at larger scales (e.g., 1B+), where model dynamics and efficiency trade-offs are known to change. This severely limits the generality and impact of the findings.

2. The paper's core premise of a GSP framework is not very strong. The authors admit that "spectral properties are not explicitly enforced" but rather "indirectly encouraged". The model does not perform operations in the spectral domain. This makes the GSP contribution feel more like a motivating analogy for a heuristic-based MoE router, rather than a principled application of GSP.

3. The method is functionally a Mixture-of-Experts (MoE) design that routes tokens to different heads. However, the paper fails to provide a direct comparison to more standard (and simpler) MoE routing mechanisms. It is unclear if the added complexity of the "spectral residual" calculation and "spectral bias" modulation offers any real advantage over a simple top-k router on the base token representations.

4.  The paper introduces several new and important hyperparameters, chief among them the ratio of shared (S) to expert (E) filters. The paper defaults to a 1:1 ratio (8 shared, 8 expert) without any ablation or justification.

**Questions:**

1. The experiments use 8 shared filters and 8 expert filters (S=8, E=8). What was the rationale for this 1:1 ratio? Could you provide any ablation studies on the effect of changing this ratio (e.g., S=4, E=12 or S=12, E=4) on performance?

2. A core motivation is that Mamba2's heads are unstructured. Did you perform a spectral analysis on the heads of a baseline Mamba2 model? it is possible that Mamba2 already learns a diverse set of low and high-pass filters, even without an explicit mechanism. Such a comparison would provide a stronger baseline.

3. Given that the GSP framework is "implicit", could you elaborate on why this GSP-inspired router is superior to a standard MoE router that simply learns to route tokens based on a linear projection of the input ?

---

> ### Author Response · Authors · 2025-11-22
> **Official Comment by Authors (1/3)**
>
> Thank you for your insightful and constructive feedback. In this response, we address each of your comments in detail and provide additional experimental results and clarifications.
>
> ## **1. Lack of scale (W1)**
>
> > **Original Comment:**
> **W1**. The paper's primary weakness is that all experiments are confined to a single, small 370M parameter model. The central claim of 58.9% parameter savings is not validated at larger scales (e.g., 1B+), where model dynamics and efficiency trade-offs are known to change. This severely limits the generality and impact of the findings.
>
>
> `Revised Section: Appendix D.2, Table 4`
>
> We have conducted additional experiments by training a 1B-parameter model on the FineWeb-Edu 30B. HADES shows strong performance against baseline models with only 71.4% of parameters.
>
>
> **Table 4. Performance comparison on language modeling and zero-shot common-sense reasoning (Appendix D.2, Table 4)**
> | Tasks | Wikitext | Lambada | Lambada |  PiQA  | hellaswag |  Wino. |  ARC_E |  ARC_C |  BoolQ | OpenbookQA |  Avg.  |
> |-|-|-|-|-|-|-|-|-|-|-|-|
> | Metric | ppl |   ppl   |   acc   |   acc  |   acc_n   |   acc  |   acc  |  acc_n |   acc  |    acc_n   |        |
> | RetNet (1.3B)|  22.45 | 21.84 |  38.70 | 69.04 |    47.73 | 52.72 | 63.68 | 33.36 | 60.61 |     36.60 | 50.31 |
> | Mamba2 (1.3B)|  **19.47** | 17.40 | 40.68   | 70.29  | **53.24**  | 56.04  | **69.87**  | **36.35**  | 55.81  | 37.40   | 52.46  |
> | DeltaNet (1.3B)|  19.77 | **16.64** | **41.78**   | 70.95  | 51.09  | 54.70  | 67.63  | 34.47  | **61.19**  | 38.40   | 52.53  |
> | *HADES* (1B) |  20.41 | 17.22 |  41.18   | **71.33**  | 51.85  | **56.35**  | 68.48  | 34.81  | 60.73  | **38.60**   | **52.92** |
>
> ---
>
> ## **2. Comparison to MoE (W3, Q3)**
>
> > **Original Comment**:
> **W3.** The method is functionally a Mixture-of-Experts (MoE) design that routes tokens to different heads. However, the paper fails to provide a direct comparison to more standard (and simpler) MoE routing mechanisms. It is unclear if the added complexity of the "spectral residual" calculation and "spectral bias" modulation offers any real advantage over a simple top-k router on the base token representations.
> **Q3**. Given that the GSP framework is "implicit", could you elaborate on why this GSP-inspired router is superior to a standard MoE router that simply learns to route tokens based on a linear projection of the input ?
>
> `Revised Section: Appendix E.1 Table 7`
>
> To address your concern, we did additional ablation study on router mechanisms. We kindly note that fixed routing and random routing were already in the manuscript. We add 3 types of routers : Softmax Top-k router and Input linear projection router. "Input-only" refers to the routing mechanism which only use input sequence to get top-ranked filters. "Gumbel Softmax Top-K" is where Top-K filters are selected with Top-K selection itself is being trained. "Weighted Aggregation" means output is aggregated via linear projection instead of simple aggregation. Our GSP router showed superior performance over both routers; indicating GSP analogy performs well. We also provide summarized table below:
>
> **Table 7.1. Ablation on routing mechanism**
> | Tasks | Wikitext | Lambada | Lambada |  PiQA  | hellaswag |  Wino. |  ARC_E |  ARC_C |  BoolQ | OpenbookQA |  Avg.  |
> |-|-|-|-|-|-|-|-|-|-|-|-|
> | Metric | ppl |   ppl   |   acc   |   acc  |   acc_n   |   acc  |   acc  |  acc_n |   acc  |    acc_n   |        |
> | Original | 31.48 | 21.74 |39.24 |63.93| 32.82 |52.64 |45.03 |22.01 |58.84 |28.80 |42.91|
> | Fixed | 34.55 | 27.64 | 35.75 | 62.40 | 31.39 | 52.88 | 44.23 | 24.23 | 60.83 | 26.00 | 42.21 |
> | Random | 35.78 | 32.77 | 33.17 | 61.97 | 30.47 | 52.49 | 43.31 | 23.12 | 55.72 | 28.00 | 41.03 |
> | Input-only (Linear routing)|	34.17|	23.95	| 38.40  | 63.71  | 31.60  | 51.22  | 43.56  | 23.04  | 58.35  | 27.80  | 42.21  |
> | Gumbel Softmax Top-K 	|34.83| 27.21 | 36.95  | 62.19  | 31.39  | 50.51  | 43.43  | 22.35  | 58.07  | 28.20  | 41.64  |
> | Weighted aggregation (MoH)	| 36.73	| 32.03 | 34.95  | 61.75  | 30.19  | 50.75  | 44.36  | 22.53  | 60.92  | 28.00  | 41.68  |

---

> ### Author Response · Authors · 2025-11-22
> **Official Comment by Authors (2/3)**
>
> ## **3. Regarding Expert/Shared Filter Ratio (W4, Q1)**
>
> > **Original Comment**:
> **W4.** The paper introduces several new and important hyperparameters, chief among them the ratio of shared (S) to expert (E) filters. The paper defaults to a 1:1 ratio (8 shared, 8 expert) without any ablation or justification.
> **Q1.** The experiments use 8 shared filters and 8 expert filters (S=8, E=8). What was the rationale for this 1:1 ratio? Could you provide any ablation studies on the effect of changing this ratio (e.g., S=4, E=12 or S=12, E=4) on performance?
>
> `Revised Section: Appendix E.2, Table 7`
>
> In the early stage of our study, we extensively explored a wide range of configurations (e.g., expert/shared filter ratio, learning rate, batch size, optimizer) while developing the core idea. Through this exploration, we found that the current configuration provided the most stable and consistent performance compared to several alternatives.
> Additional ablation studies further confirm this observation, demonstrating that the adopted setting yields the best performance among the tested options.
>
> We initially chose the 1:1 ratio (S=8, E=8) as it performed well when we experimented simple filter selection without bias on Mamba2 model. We also note that the two extreme cases (S=16, E=0 and S=0, E=16) are already included in the manuscript. To address your concern, we conducted additional ablations with S=4, E=12 (25%) and S=12, E=4 (75%), while keeping the total number of filters fixed. The full results are included in the revised manuscript (Appendix E.2, Table 7), and we also provide them below:
>
> **Table 7.2. Ablation on filter selection ratio**
> | Tasks | Wikitext | Lambada | Lambada |  PiQA  | hellaswag |  Wino. |  ARC_E |  ARC_C |  BoolQ | OpenbookQA |  Avg.  |
> |-|-|-|-|-|-|-|-|-|-|-|-|
> | Metric | ppl |   ppl   |   acc   |   acc  |   acc_n   |   acc  |   acc  |  acc_n |   acc  |    acc_n   |        |
> | Filter 16 / 0 (0%) | 36.34 | 30.12 | 34.89 | 61.53 | 30.30 | 52.41 | 44.49 | 22.70 | 58.29 | 28.80 | 42.21 |
> | Filter 16 / 4 (25%) |35.24 | 29.53 | 34.64  | 62.79  | 30.89  | 50.43  | 43.35  | 23.81  | 59.48  | 28.20 | 41.70 |
> | Filter 16 / 8 (50%) | 31.48 | 21.74 |39.24 |63.93| 32.82 |52.64 |45.03 |22.01 |58.84 |28.80 |42.91|
> | Filter 16 / 12 (75%) | 35.92 | 31.67 |34.08  | 62.35  | 30.12  | 50.83  | 42.89  | 23.21  | 61.80  | 28.40 | 41.71 |
> | Filter 16 / 16 (100%) | 34.55 | 27.64 | 35.75 | 62.40 | 31.39 | 52.88 | 44.23 | 24.23 | 60.83 | 26.00 | 41.68 |

---

> ### Author Response · Authors · 2025-11-22
> **Official Comment by Authors (3/3)**
>
> ## **4. Spectral Analysis of Mamba2 Head (Q2)**
>
> > **Original Comment**:
> **Q2.** A core motivation is that Mamba2's heads are unstructured. Did you perform a spectral analysis on the heads of a baseline Mamba2 model? it is possible that Mamba2 already learns a diverse set of low and high-pass filters, even without an explicit mechanism. Such a comparison would provide a stronger baseline.
>
> `Revised Section: Section 1, Fig. 1, Section 4, Fig. 5 and Fig. 6, Appendix E.3, Fig. 8`
>
> We thank the reviewer for pointing out that vanilla Mamba2 heads may already exhibit diversity in their dynamic filtering behavior. We believe it would be better to show filter response  analysis (Section 4, Fig. 6) with effective rank (Section 1, Fig 1) in main paper. Also, we conducted extended analyses on filter outputs; spectrum analysis (Section 4, Fig. 5) and head-wise CKA (Appendix E.3, Fig. 8).
>
> Across filter response analysis in Fig. 6, we consistently observe that a large fraction of Mamba2’s learned heads behave as near smooth kernel, leads Mamba2 to primarily capture low-frequency or globally smooth information while failing to represent high-frequency details. As a result, many heads converge to similar general-purpose behaviors. The CKA heatmap in Fig. 8 further shows that Mamba2 contains many redundant filters, with numerous filter pairs exhibiting high similarity.
>
> This phenomenon aligns with the low effective rank reported in Fig. 1, indicating that many heads operate redundantly in overlapping spectral regimes rather than forming a diverse filter bank. This makes clear why HADES introduces selection and modulation structure. This structure follows directly from the spectral decomposition suggested by our GSP view, even though we do not explicitly use GSP operators for efficiency reasons.
>
> The revised figures show that this design choice is realized during training. As shown in Fig. 6, HADES learns both smooth and ripple-shaped kernels; leverages not only low-frequency dominant global context but also high-frequency driven fine-grained structure in a more balanced manner. This indicates that our high-frequency–aware modulation and routing design enables the model to capture a wider range of spectral patterns. Correspondingly, HADES achieves a substantially higher effective rank (Fig. 1) and exhibits far less redundancy in the CKA structure (Fig. 8, Center figures). Moreover, the output spectra in Fig. 5 (b) and (c) reflect this specialization: shared filters capture stable, low-frequency content, while expert filters—with ripple kernels—focus on sharper, token-specific variations. These results demonstrate that the GSP-inspired principles not only motivate the architecture but also emerge clearly in the learned filter dynamics.
>
> ---
>
> ## **5. GSP Analogy (W2)**
>
> > **Original Comment**:
> **W2.** The paper's core premise of a GSP framework is not very strong. The authors admit that "spectral properties are not explicitly enforced" but rather "indirectly encouraged". The model does not perform operations in the spectral domain. This makes the GSP contribution feel more like a motivating analogy for a heuristic-based MoE router, rather than a principled application of GSP.
>
> `Revised Section: Section 1, Fig.1 / Section 4, Fig. 5 and Fig. 6 / Appendix E.3, Fig. 8`
>
> We agree that our model does not explicitly operate in the spectral domain. Our GSP perspective is implicit rather than enforced through explicit spectral operators. However, it is more than just a retrospective analogy. The spectral analyses discussed in previous question—effective rank, filter response, output spectrum and CKA—were part of the original set of diagnostic tools that motivated the architecture, and we expanded their presentation for clarity in the revision.
>
> These analyses show that Mamba2 may have meaningless filters, without forming meaningful spectral decomposition across heads. This aligns naturally with classical GSP insights, where explicit separation of smooth (low-frequency) and residual (high-frequency) components often leads to more structured processing. Guided by this intuition, we introduced the spectral-residual term $r_t$ to emphasize high-frequency content and to control both expert selection and $\Delta$-adjustment.
>
> While we do not introduce explicit DSP operators for efficiency reasons, this spectral intuition guided the mechanism we propose. Importantly, the trained filters confirm that the intended spectral structure emerges, supporting the validity of this implicit GSP-inspired design. Thus, while our GSP contribution is implicit, it meaningfully shaped both the design and the resulting learned dynamics.

---

> > ### Author Response · Authors · 2025-11-26
> >
> > Dear Reviewer kLjo,
> >
> > Thank you very much for the time and care you devoted to reviewing our work. We truly appreciate your insightful feedback, which has been invaluable in helping us improve the paper. During the discussion period, we carefully addressed each of your comments in our responses, and we kindly ask you to review them and reconsider your rating if you find our clarifications satisfactory.
> >
> > We are grateful for your thoughtful evaluation, and please feel free to reach out if any additional questions arise.
> >
> > Best regards,
> > Authors

---

### Author Response · Authors · 2025-11-22
**Notice of Updates**

We thank all reviewers for their constructive comments. For clarity, the table and figure numbers referenced in this rebuttal are aligned with those in the revised manuscript. We have uploaded the updated paper accordingly, with the following major changes.

* Section 1: Update Introduction and add figure of effective rank analysis (Fig.1)
* Section 2.1: Minor correction on SSM equation
* Section 3.1: Minor clarification on $\mu_t$
* Section 4.1: Update Table 1 with adding train perplexity of all models
* Section 4.2: Update analysis on filter response and output spectrum. Added figures and rearranged existing ones.(Fig.5, Fig.6, and Fig.7)
* Section 4.3: Update Table 2 with adding more ablation studies
* Appendix: Major rearrangement of Appendix.
    * Appendix C. Extended Experiments to Appendix D. Detailed Benchmarks and More Evaulations, Appendix E. Extended Model Analyses and Appendix F. Efficiency and Computational Complexity.
    * Also Rearranged Appendix D. More Discussion to Appendix B. More Discussion
* Appendix C.1: Minor clarification on training details
* Appendix D.2: Add section on 1B parameter scale experiment (Table 4) and comparison to mixture variant (Table 5)
* Appendix E.1: Add more ablations on routing and filter ratio (Table 6)
* Appendix E.3: Add CKA analysis (Fig.8)
* Appendix E.4: Update analysis details on frequency spectrum analysis of input and output sequence, rearranged figure. (Fig.9)
* Appendix E.5: Add analysis details on frequency spectrum analysis of filter response and additional figure (Fig.10)
* Appendix F.1: Add details on parameter size and 370M result, minor correction on setup
* Appendix F.4: Minor calculation fix
* Appendix F.5: Add FLOPs analysis (Table 13 and 14)
* Appendix F.6: Add Training complexity analysis (Fig.11)
* Appendix G: Update limitation
* Appendix H: Add visualizations regarding difference of $\gamma$-value (Fig. 12)

---

### Author Response · Authors · 2025-11-29
**Summary of Rebuttal Discussion for New Area Chair**

For your convenience, we briefly summarize the earlier discussion during the rebuttal phase. We appreciate the reviewers’ engagement, and note that the initial scores of **(4, 6, 4, 6)** were updated to **(4, 6, 6, 6)** before being rolled back due to an anonymous leak. We provide summary of rebuttal for each reviewers below:

* **Reviewer kLjo (4 $\rightarrow$ no response yet)**
    *  Concerns and Our Responses
        *  Scalability of experiments $\rightarrow$ Provided 1B-scale experiments demonstrating consistent performance.
        *  Analysis on Mamba2 and Connection to GSP $\rightarrow$ Added extended spectral analysis clarifying the conceptual link between GSP and HADES.
        *  Ablation of routing and filter selection ratio $\rightarrow$ Included targeted ablations as requested.
* **Reviewer QPLB (6 $\rightarrow$ no response yet)**
    * Concerns and Our Responses
        * FLOPs overhead of HADES $\rightarrow$ Delivered FLOPs analysis showing routing overhead is minimal relative to performance gains.
        * Confound spectral analysis $\rightarrow$ Expanded and clarified spectral analysis to resolve misunderstandings.
        * Ablation of routing mechanism $\rightarrow$ Added ablation results examining routing behavior.
        * Experimental details (passkey, running mean) $\rightarrow$ Provided detailed explanations with pseudocode.
* **Reviewer neR9 (4 $\rightarrow$ 6)**
    * Concerns and Our Responses
        * Scalability and robustness of HADES $\rightarrow$ Added 1B-scale results, seed sweep, parameter-matching experiments, and train-perplexity reports.
        * Architectural questions (Top-K, load-balance loss) $\rightarrow$ Provided clarifications with pseudocode.
        * The reviewer acknowledged the clarifications and raised the score from 4 to 6.
* **Reviewer BvVa (6 $\rightarrow$ no response yet)**
    * Concerns and Our Responses
        *  Connection to GSP $\rightarrow$ Added extended spectral analysis explaining how the GSP analogy relates to HADES.
        *  FLOPs overhead of HADES $\rightarrow$ Included FLOPs analysis demonstrating minimal overhead compared to reduction of FLOPs through selection.
        *  Comparison to mixture models $\rightarrow$ Added direct comparisons showing HADES outperforms mixture-based alternative.
        *  Experimental and architectural questions (Top-K, training behavior) $\rightarrow$ Provided additional architectural details and training-behavior analysis.

We would like to express our sincere appreciation to the Area Chair for the careful handling of our submission under an unusual review circumstance. We hope that the additional analyses, clarifications, and experiments we provided help convey the technical soundness and contributions of our work. Thank you for your time and thoughtful consideration.

Best regards,

HADES Authors

---

### Meta-Review · Area_Chair_6kGe · 2026-01-11

**Summary:**

The paper proposes a new framework called HADES (Hierarchical Adaptive filter bank for Efficient SSMs) inspired by Graph Signal Processing (GSP) that reinterprets the Mamba2 architecture as an adaptive filter bank on a line graph. The authors introduce a hierarchical structure consisting of shared filters for global low-pass information and expert filters for local high-pass details, selected via a spectral residual-based routing mechanism. The primary claim is that HADES achieves comparable performance to Mamba2 while utilizing significantly fewer parameters (approx. 59% of the original).

The model demonstrates impressive parameter efficiency, maintaining competitive performance on language modeling and reasoning tasks with a reduced parameter count. The GSP-inspired lens provides a more structured and potentially interpretable view of head-wise recurrence in SSMs compared to the unconstrained heads in vanilla Mamba2.

The authors significantly strengthened the paper by adding FLOPs analysis, spectral response visualizations, and more comprehensive benchmarks during the discussion period. A major concern regarding the initial 370M model scale was effectively addressed during the rebuttal. The authors provided new results for a 1B-parameter model on the FineWeb-Edu dataset, showing consistent performance gains over baselines like Mamba2 and DeltaNet.



On the other hand, there are the following concerns on this paper:
- Gap between Theory and Practice: While the GSP framework is an interesting motivating analogy, the connection remains somewhat superficial. The mathematical derivations of line graph filters are largely equivalent to well-known 1-D convolutions. The actual implemented algorithm and the spectral residual-based routing appear more heuristic than directly derived from a rigorous GSP theorem.
- Insufficient Baseline Comparisons: The comparison against more standard Mixture-of-Experts (MoE) or mixture-based SSMs was initially weak. While the authors added a comparison with MoM (Mixture-of-Mamba) in the rebuttal, it is still unclear whether the proposed GSP method is significantly superior to those rather naive approaches.
- Initial Ablation Gaps: The original submission lacked sufficient ablation studies on key hyperparameters like the shared-to-expert filter ratio. Although addressed in the rebuttal with a 1:1 ratio justification and additional sweeps, these details are crucial for the main body of the paper.



As a summary, this is a borderline paper where the empirical results and the authors' proactive rebuttal outweigh the theoretical weaknesses. The theoretical contribution is admittedly incremental and acts more as a diagnostic lens than a rigorous generative theory. However, the achievement of competitive performance with nearly 40% fewer parameters at the 1B scale is a significant practical result for the SSM community. The authors have addressed the reviewers' primary concerns during the rebuttal period. For these reasons, AC recommends acceptance.

**Reviewer Concerns:**

I did not find any reviewer concerns. All reviewers provided appropriate reviews.

**Reviewer Scores:**

As Reviewer neR9 stated in the discussion, they increased the score to 6.
Reviewer kLjo would have raised their score to 6, because their concerns were addressed appropriately.

---

### Decision · Program_Chairs · 2026-01-26

Accept (Poster)